# Detecting high spatial variability of ice-shelf basal mass balance, Roi Baudouin ice shelf, Antarctica

Sophie Berger[1], Reinhard Drews[2], Veit Helm[3], Sainan Sun[1], and Frank Pattyn[1]

[1]Laboratoire de glaciologie, Université libre de Bruxelles, Brussels, Belgium
[2]Department of Geosciences, University of Tübingen, Tübingen, Germany
[3]Alfred Wegener Institute, Bremerhaven, Germany

*Correspondence to:* S. Berger (sberger@ulb.ac.be)

**Abstract.**

Ice shelves control the dynamic mass loss of ice sheets through buttressing and their integrity depends on the spatial variability of their basal mass balance (BMB), i.e., the difference between refreezing and melting. Here, we present an improved technique – based on satellite observations – to capture the small-scale variability in the BMB of ice-shelves. As a case study, we apply the methodology to the Roi Baudouin Ice Shelf, Dronning Maud Land, East Antarctica and derive its yearly-averaged BMB at 10 m horizontal gridding. We use mass conservation in a Lagrangian framework based on high-resolution surface velocities, atmospheric-model surface mass balance and hydrostatic ice-thickness fields (derived from TanDEM-X surface elevation). Spatial derivatives are implemented using the total-variation differentiation, which preserves abrupt changes in flow velocities and their spatial gradients. Such changes may reflect a dynamic response to localised basal melting and should be included in the mass budget. Our BMB field exhibits much spatial detail and ranges from -14.7 to 8.6 m a$^{-1}$ ice equivalent. Highest melt rates are found close to the grounding line where the pressure melting point is high, and the ice-shelf slope is steep. The BMB field agrees well with on-site measurements from phase-sensitive radar, although independent radar profiling indicates unresolved spatial variations in firn density. We show that an elliptical surface depression (10 m deep and with an extent of 0.7×1.3 km) lowers by 0.5 to 1.4 m a$^{-1}$, which we tentatively attribute to a transient adaptation to hydrostatic equilibrium. We find evidence for elevated melting beneath ice-shelf channels (with melting being concentrated on the channel's flanks). However, farther downstream from the grounding line, the majority of ice-shelf channels advect passively (i.e. no melting nor refreezing) toward the ice-shelf front. Although the absolute, satellite-based BMB values remain uncertain, we have high confidence in the spatial variability on sub-kilometre scales. This study highlights expected challenges for a full coupling between ice and ocean models.

## 1 Introduction

Approximately 74% of the Antarctic ice sheet is surrounded by floating ice shelves (Bindschadler et al., 2011a) providing the interface for interactions between ice and ocean. Marine ice sheets – characterized by a bed elevation below sea level and sloping down towards the interior – can be destabilised leading to a marine ice sheet instability (Mercer, 1978; Schoof, 2007; Tsai et al., 2015). However, ice shelves that are laterally constrained through embayments (or locally regrounded from below),

mitigate the marine ice sheet instability (Gudmundsson et al., 2012), thus regulating the ice flux from the inland ice sheet through buttressing. Over the last decade, major advances in our understanding of the processes at this ice-ocean interface have emerged, both theoretically (e.g. Pattyn et al., 2013; Favier et al., 2012, 2014; Ritz et al., 2015) as well as from observations (e.g. Rignot et al., 2014; Wouters et al., 2015). It is now established that ice-shelf integrity plays an important part in explaining sea-level variations in the past (Golledge et al., 2014; DeConto and Pollard, 2016), enabling improved projections of future sea-level rise (Golledge et al., 2015; Ritz et al., 2015).

Ice-shelf integrity can be compromised by atmospheric driven surface melt-ponding (Lenaerts et al., 2017) and hydrofracturing (Banwell et al., 2013; Scambos et al., 2004; Hulbe et al., 2004). From the ocean side, ice shelves may thin or thicken (Paolo et al., 2015) due to changes in basal mass balance (BMB), i.e. the difference between refreezing and melting. Point measurements with phase-sensitive radars (Marsh et al., 2016; Nicholls et al., 2015), global navigation satellite system (GNSS) receivers (Shean et al., 2017), observations from underwater vehicles (Dutrieux et al., 2014) and analysis from high-resolution satellites (Dutrieux et al., 2013; Wilson et al., 2017) have shown that BMB varies spatially on sub-kilometre scales. Ice-shelf channels are one expression of localised basal melting (Stanton et al., 2013; Marsh et al., 2016) which, after hydrostatic adjustment, form curvilinear depressions visible at the ice-shelf surface (Fig. 1). These surface depressions reflect basal incisions resulting in curvilinear tracts of thin ice. In some areas, ice-shelf channels are twice as thin as their surroundings (Drews, 2015). However, the impact of ice-shelf channels on ice-shelf integrity is yet unclear because, on the one hand, excessive basal melting beneath ice-shelf channels may prevent ice-shelf-wide thinning (Gladish et al., 2012; Millgate et al., 2013) but, on the other hand, increased crevassing due to channel carving may structurally weaken the ice shelf (Vaughan et al., 2012).

Here we attempt to derive the BMB of the Roi Baudouin Ice Shelf (RBIS), Dronning Maud Land, East Antarctica, at 10 m gridding, based on mass conservation in a Lagrangian framework. The RBIS (Fig.1) is constrained by an ice promontory to the West and by Derwael Ice Rise in the East, blocking the tributary flow from Western Ragnhild Glacier – one of the largest outlet glaciers in Dronning Maud Land (Callens et al., 2014). Analyses on Derwael Ice Rise (Drews et al., 2015; Callens et al., 2016) and the larger catchment area (Favier et al., 2016) suggest that the RBIS is a relatively stable sheet-shelf system on millennial time scales. The RBIS contains a number of ice-shelf channels (Drews, 2015, and arrows in Fig. 2e), many of which start at the grounding line and extend over 230 km to the ice-shelf front.

We outline our approach of deriving the BMB, with focus on attaining high spatial resolution. Resolving BMB is challenging, because it is computed as the residual of several uncertain quantities and it relies on spatial derivatives, which amplify noise in the input data. The latter can be accounted for with spatial averaging (e.g. Neckel et al., 2012; Moholdt et al., 2014), which, however, may smear out the imprint of processes acting on sub-kilometre scales. Here, we use spatially well-resolved input data combined with total-variation regularization of the velocity gradients. This avoids spatial averaging, but still mitigates the noise in the input data. As a result, our BMB field shows high detail over different spatial scales that are validated with phase-sensitive radar, GNSS observations and ground-penetrating radar.

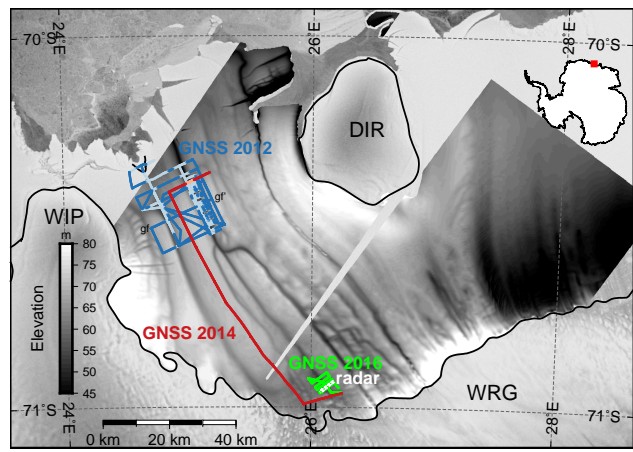

**Figure 1.** Overview of the topography (grey shading) of the Roi Baudouin Ice Shelf (from TanDEM-X 2014) and ground-truth datasets presented and discussed in the text (Sect 2.7 and 4). Acronyms stand for DIR: Derwael Ice Rise, WIP : Western Ice Promontory and WRG: West Ragnhild Glacier. The profile gf-gf' is shown in Fig. S2. Light blue and light red are the low-lying parts of the ice-shelf, which are excluded from the GNSS-TanDEM-X comparison in Fig. 9. "Radar" denotes both ground-penetrating and phase sensitive radars. The background is from the Radarsat mosaic (Jezek and RAMP-Product-Team, 2002) and the black line delineates the grounding line (Depoorter et al., 2013).

## 2 Data and methods

### 2.1 Basal mass balance from mass conservation

We derive the basal mass balance ($\dot{M}_b$) from mass conservation, i.e.,

$$\dot{M}_b = \frac{\partial H_i}{\partial t} + \nabla \cdot (H_i \boldsymbol{u}) - \dot{M}_s \tag{1a}$$

$$= \frac{\partial H_i}{\partial t} + (\boldsymbol{u} \cdot \nabla H_i + H_i \nabla \cdot \boldsymbol{u}) - \dot{M}_s \tag{1b}$$

$$= \frac{\mathrm{D}H_i}{\mathrm{D}t} + H_i (\nabla \cdot \boldsymbol{u}) - \dot{M}_s \tag{1c}$$

where $\dot{M}_s$ is the surface mass balance (SMB, positive values for mass gain), $H_i$ is the ice thickness and $\boldsymbol{u}$ the column-average horizontal velocity of the ice. $\dot{M}_b$, $\dot{M}_s$ and $H_i$ are given in ice-equivalent units. $\partial H_i / \partial t$ and $\mathrm{D}H_i / \mathrm{D}t$ represent the Eulerian and Lagrangian thickness change, respectively and $\nabla \cdot (H_i \boldsymbol{u})$ denotes the flux divergence, that includes advection of thickness gradients ($\boldsymbol{u} \cdot \nabla H$) and ice-flow divergence ($H_i \nabla \cdot \boldsymbol{u}$). In principle, Eq. (1) does not depend on the reference frame and can be calculated in both a fixed coordinate system (i.e. Euler coordinates) or with a moving coordinate system that follows the ice flow (i.e. Lagrange coordinates). In practice, however, both approaches differ: Eulerian studies are often based on one thickness field and either assume steady-state (Rignot and Steffen, 2008; Neckel et al., 2012) or rely on an external dataset (Depoorter et al., 2013; Rignot et al., 2013) to account for the thickness changes $\partial H_i / \partial t$ (e.g. Pritchard et al., 2012; Paolo et al., 2015).

**Table 1.** Key features of the main input datasets used to compute the variables of Eq. (1c).

| Type of data | Observations/Modelling Reference | Dataset/Model | gridding | Use (Eq.) | Average (min ; max) |
|---|---|---|---|---|---|
| Surface elevation | **Observations** this study | TanDEM-X | 10 m | $\frac{\mathrm{D}H_i}{\mathrm{D}t}, H_i$ | 63.8 m (19.8 ; 117.4) |
| Velocity | **Observations** Berger et al. (2016) | ERS1/2 ALOS PALSAR | 125 m | $\nabla \cdot \boldsymbol{u}$ | 189.7 m a$^{-1}$ (0.1 ; 378.2) |
| Surface Mass Balance | **Modelling** Lenaerts et al. (2017, 2014) | RACMO 2.3 | 5.5 km | $\dot{M}_s$ | 0.3 m a$^{-1}$ (0.0 ; 1.0) |
| Firn-air content | **Modelling** Lenaerts et al. (2017) Ligtenberg et al. (2011) | RACMO 2.3 | 5.5 km | $H_i$ | 12.8 m (0.0 ; 22.5) |
| Mean Dynamic topography | **Modelling & observations** Knudsen and Andersen (2012) | DTU12MDT | 0.125° | $H_i$ | -0.1 m (-0.9 ; 0.6) |
| Geoid | **Modelling & observations** Förste et al. (2014) | EIGEN-6C4 | 0.125° | $H_i$ | 17.0 m (14.6 ; 19.8) |

The Lagrangian approach, on the other hand, requires two thickness fields covering different time periods from which the Lagrangian thickness change is calculated implicitly ($\mathrm{D}H_i/\mathrm{D}t$). As shown below, the key difference between both approaches is how the advection of thickness gradients ($\boldsymbol{u} \cdot \nabla H$) is accounted for. The Lagrangian approach is best-suited in areas with rough surface and significant advection (e.g. near ice-shelf channels). We refer to previous publications (Dutrieux et al., 2013; Moholdt et al., 2014; Shean et al., 2017) that further explain differences between Eulerian and Lagrangian approaches.

In the following, we describe surface velocities in Sect. 2.2, surface mass balance in Sect. 2.3, the derivation of hydrostatic ice thickness in Sect. 2.4 and Lagrangian thickness change in Sect. 2.5. Key features of the input datasets are summarised in Table1. As a novelty compared to previous studies, we base our hydrostatic thickness field on high-resolution digital elevation models (DEMs) derived from TanDEM-X images from 2013 and 2014. Section 2.6 explains the implementation of spatial velocity gradients ($\nabla \cdot \boldsymbol{u}$ in Eq. (1)), which is non-trivial when derivatives are taken over short distances with noisy input data. We compare the derived Lagrangian Basal Mass Balance (LBMB) with field measurements of phase-sensitive radar and GNSS profiling (Sect. 2.7). Although this is not a direct validation, as the field data cover a different period, the comparison is insightful to understand the spatial variability in our LBMB field. The derived LBMB is only valid in freely floating areas, which excludes the grounding zone, but also other small-scale features such as ice-shelf channels where viscous inflow can occur (Humbert et al., 2015; Drews, 2015). (Examples where this may be the case are discussed in Sect. 5).

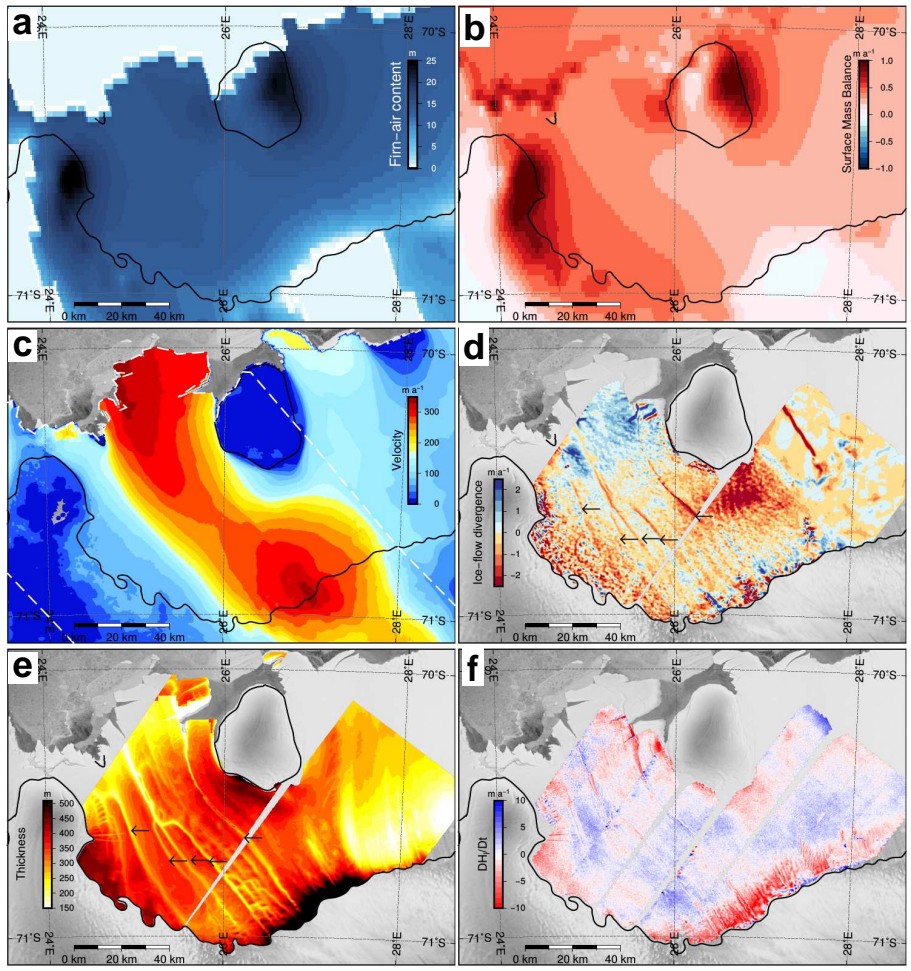

**Figure 2.** (a, c, e) Variables entering Eqs. (1) and (2) and (b, d , f) terms needed to calculate the LBMB in Eq. (1).

(a) Firn-air content ($H_a$); (b) surface mass balance ($\dot{M}_s$); (c) surface velocities ($\boldsymbol{u}$) (the white dashed line delinates the flow field from Berger et al. (2016)); (d) ice-flow divergence ($H_i\left(\nabla \cdot \boldsymbol{u}\right)$) (Note: the red band 30km east of Derwael ice rise is caused by a seam in Rignot et al. (2011b)'s flow field); (e) hydrostatic ice thickness of 2014 ($H_i$) and (f) Lagrangian thickness change ($\mathrm{D}H_i/\mathrm{D}t$). Arrows in (d) and (e) locate ice-shelf channels. The background is from the Radarsat mosaic (Jezek and RAMP-Product-Team, 2002) and the black line delineates the grounding line (Depoorter et al., 2013). Key features regarding the input datasets are summarised in Table 1.

## 2.2 Surface velocities from satellite radar remote sensing

Assuming that velocities do not vary with depth, we use surface velocities that were previously derived by combining interferometric Synthetic Aperture Radar (InSAR) and speckle tracking (Berger et al., 2016). The velocities are mosaicked and gridded to a 125 m posting and are based on images from the European Remote Sensing satellites (ERS 1/2) from 1996 and the Advanced Land Observing System Phased Array type L-band Synthetic Aperture Radar (ALOS-PALSAR) from 2010. As shown in Berger et al. (2016), comparison with on-site measurements collected in 1965-1967 and 2012-2014 yields no evidence of prominent changes in the ice velocities over the last decades, which supports the combination of data from different dates. The velocity mosaic covers 75% of our area of interest (dashed line in Fig. 2c). The remaining areas are filled with an Antarctic-wide flow field (Rignot et al., 2011b) gridded to 900 m postings (the 450 m gridded velocities are too noisy in our area of interest). We reduce seams – as high as 60 m a$^{-1}$ in some places – using linear feathering (e.g. Joughin, 2002; Neckel et al., 2012) over 4.5 km.

## 2.3 Surface mass balance from atmospheric modelling

We use the surface mass balance from a high-resolution (5.5 km posting) simulation of the Regional Atmospheric Climate MOdel (RACMO) version 2.3, centred on Dronning Maud Land ($25\,°$W and $45\,°$W) and averaged over the period 1979–2015 (Lenaerts et al., 2014, 2017). The SMB field correctly reproduces asymmetries across Derwael Ice Rise originating from orographic uplift and also simulates a corresponding shadowing effect on the Roi Baudouin Ice Shelf (Fig. 2b and Lenaerts et al., 2014). Moreover, the simulation explains observed surface melting near the grounding zone due to a wind-albedo feedback caused by persistent katabatic winds in this area (Lenaerts et al., 2017).

## 2.4 Hydrostatic Ice Thickness

We calculate the ice thickness (Fig. 2e) by imposing hydrostatic equilibrium on surface freeboard (Bindschadler et al., 2011b; Chuter and Bamber, 2015; Drews, 2015) derived from the TanDEM-X satellites. The details of hydrostatic inversion are presented in the following two sections.

### Surface elevations

The digital elevation models are processed from 43 image pairs (Fig. S1) of the TanDEM-X mission (Krieger et al., 2007), in which the TerraSAR-X and TanDEM-X satellites image the surface simultaneously from different viewing angles. This allows to infer topography interferometrically without the need to correct for ice flow. Images from the austral winters of 2013 and 2014 are processed to single-look complex scenes, using SARscape®. After coregistration using the CryoSat-2 DEM (Helm et al., 2014), the pairs are differenced in phase. The resulting interferograms are then unwrapped and the phase difference is re-flattened before being geo-referenced in polar stereographic coordinates. The processing provides 43 single DEMs (32 from 2013 and 11 from 2014) gridded to 10 m. They cover time spans of June-October 2013 and June-July 2014 (Fig. S1), with time separations ranging from 231 to 379 days in areas where thickness rates where calculated between the two years.

Digital elevation models from the same date and satellite path are concatenated together, with a linear taper on overlapping zones. Grounded areas are masked out using the composite grounding line from Depoorter et al. (2013), based on differential InSAR with Radarsat and PALSAR (Rignot et al., 2011a) at RBIS. To correct for small elevation shifts between the different frames, which we assume to be uniform over the ice shelf, we tie the 2013 concatenated frames to each other and to the CryoSat-2 DEM (Helm et al., 2014), using constant offsets. We attribute these small shifts to tides, inverse barometric effects or different calibrations during the SAR processing.

All DEMs are smoothed with a Gaussian filter to remove small-scale surface roughness. The standard deviation of the filter is set to 7 pixels (or 70 m) in all directions. This means that points lying within that distance are weighted with 0.68. To determine the size of the Gaussian filter, we investigated standard deviations from 1-10 pixels and found that using 7 pixels minimises the elevation discrepancy between 2012 GNSS and TanDEM-X surface elevation (Sect. 2.7). As shown in Fig. S2, the applied smoothing does not affect the shape of the surface depressions linked to ice-shelf channels (with a typical width of 1-2 km).

The difference fields of the 2013-2014 overlapping DEMs exhibit a linear trend aligned with the satellite trajectory. We attribute this signal to the interferometric processing, which can leave a flawed elevation trend due to imprecise information about the satellite orbits or due to ill-constrained parameters during the SAR processing (Drews et al., 2009). To account for this effect, we subtract a plane from the 2014 DEM using the difference fields of 2013-2014 overlapping fields. The plane fit and the offset correction applied earlier mask absolute $\partial H_i/\partial t$ changes, which we assume to be small in the following.

To assess the relative vertical accuracy of the final DEMs (Sect. 4.1) (i) we use the difference fields of overlapping, un-concatenated TanDEM-X frames from the same date and satellite path (Fig. S3), and (ii) we compare the DEMs to kinematic GNSS profiling. We estimate the relative vertical accuracy to be better than 1 m, although elevation differences in some areas are systematically higher (Sect 2.7). The offset and plane fitting corrections are further discussed in Sect. 4.1, as they strongly impact the quality of our ice-thickness fields and the resulting LBMB rates.

**Hydrostatic equilibrium**

We invert hydrostatic thickness from freeboard heights ($h_{asl}$) with densities of $\rho_w$=1027 kg m$^{-3}$, $\rho_i$=910 kg m$^{-3}$ and $\rho_a$=2 kg m$^{-3}$, for seawater, ice and firn air, respectively :

$$H_i = \frac{\rho_w h_{asl}}{\rho_w - \rho_i} - \frac{H_a(\rho_w - \rho_a)}{\rho_w - \rho_i}. \tag{2}$$

The firn-air content $H_a$ accounts for the lower firn and snow densities by subdividing the ice column in air- and ice-equivalent layers. We use simulated values from the firn-densification model 'IMAU-FDM' (Fig. 2a and Ligtenberg et al., 2011; Lenaerts et al., 2017), which is forced by the SMB, exists on the same spatial grid (5.5 km, Sect. 2.3) and is averaged over the same time-period (1979-2015). For converting ellipsoidal heights to freeboard elevations we employ the EIGEN-6C4 geoid (Förste et al., 2014) and the DTU12MDT mean dynamic topography model (Knudsen and Andersen, 2012). The hydrostatic ice thickness is most sensitive to the firn-air content and the freeboard heights, resulting in an estimated uncertainty of at least $\pm$ 25 m (Drews, 2015). However, as discussed in Sect. 4.1, uncertainties can be much higher in areas where firn density is ill-constrained.

## 2.5 Lagrangian thickness change

As the Lagrangian framework moves with the flow, computing the Lagrangian thickness change $\mathrm{D}H_i/\mathrm{D}t$ requires to shift one thickness field to match the geometry of the second one. Consequently, this approach implicitly accounts for advection of thickness gradients ($\boldsymbol{u} \cdot \nabla H_i$). Here, the 2013 TanDEM-X frames are shifted forward with a normalized correlation-coefficient

matching algorithm from the computer vision library OpenCV (Bradski and Kaehler, 2008). Each 2013 concatenated frame is divided in 5×5 km patches that are sampled every kilometre in both directions. Each 2013 patch is then compared with any possible 5×5 km patch within a slightly bigger search region (6.6×6.6 km) in the 2014 DEMs that overlap with the 2013 DEM. Comparison is based on normalized cross-correlation coefficients technique, a more robust variant of 2D normalised cross correlation (Marengoni and Stringhini, 2011). The shift of the 2013 patches is found where the correlation coefficient

is maximal. Mismatches are discarded when the correlation-coefficient is smaller than 0.8, or when the detected offset is well beyond what would be expected from the available flow-field. All the 2013 shifted patches are then mosaicked to construct a shifted 2013 frame that matches the geometry of its overlapping 2014 frame. The process is applied to each overlapping pair of 2013-2014 TandDEM-X frames before conversion to hydrostatic thickness.

In Sect. 4.1, we investigate an alternative approach using observed surface velocities to shift the DEMs with a 10 day time-

step (as in Moholdt et al., 2014). We also apply this alternative approach to shift the 2016 GNSS profiles (Sects. 2.7, 3.2 and 5).

## 2.6 Spatial derivatives of noisy input data

Taking spatial gradients in Eq. (1) is not straightforward as naive discretization schemes (e.g. forward, backward or central differences) greatly amplify the signal-to-noise ratio if the input data are noisy. This issue can be accounted for by smoothing

the input data (e.g. Moholdt et al., 2014) and/or by increasing the lateral distances over which the derivative is approximated (e.g. Neckel et al., 2012). However, smoothing prior to taking the derivative can lead to smearing out of the derivative in areas where the derived quantity changes abruptly (or discontinuously). We expect such abrupt changes in the surface velocities across ice-shelf channels that experience strong basal melting (Drews, 2015). To circumnavigate this problem, we applied the total-variation regularization, a technique that suppresses noise from spatial derivatives while preserving abrupt changes

(Chartrand, 2011). Noise is removed from the data by reducing the total variation of the signal to a certain degree controlled by a regularization parameter $\alpha$ (Chartrand, 2011). The $\alpha$ value we use ($10^5$) is given by the variance of the velocities, following the discrepancy principle (Chartrand, 2011). Figure 3 compares regularized derivatives with derivatives based on velocity fields that were smoothed to varying degrees prior to taking the derivatives using central differences. It should be noted that some ambiguity about the specific choice of $\alpha$ remains but this is inherent to regularization in general. We discuss the benefits and

trade-offs of the different derivative schemes further in Sect. 4.2.

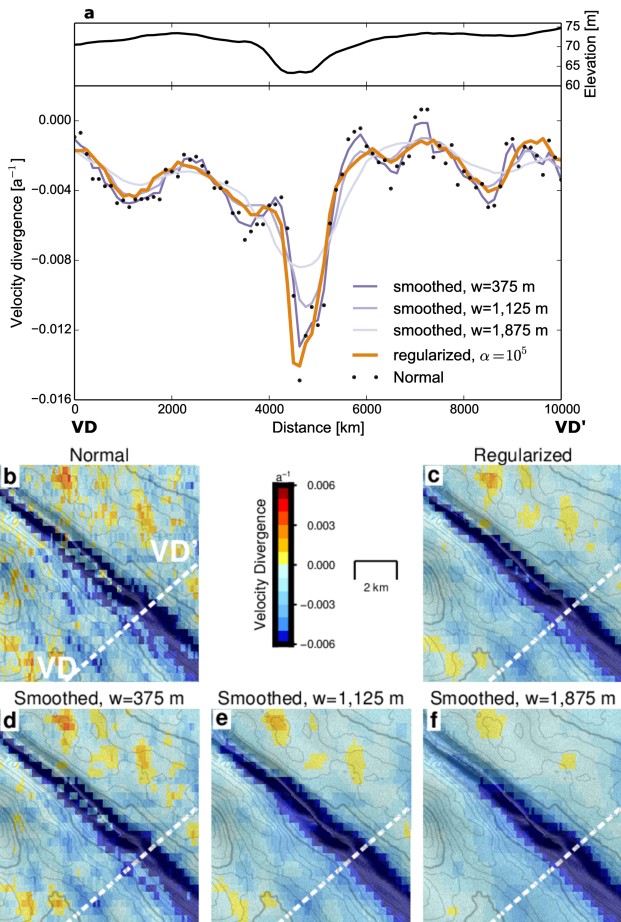

**Figure 3.** Velocity divergence at an ice-shelf channel located in Fig 4. (a) Profile showing elevation and velocity divergence for various degree of smoothing (w = window width) and after regularization ($\alpha = 10^5$). (b -f) corresponding spatial pattern of the velocity-divergence profiles shown in (a) . The background image is from Landsat 8, acquired in 2014 and the maps are overlain with elevation contour lines of 1 m.

## 2.7 On-site geophysical measurements

Remote-sensing and modelling data are complemented by a series of geophysical measurements (ground-penetrating radar, GNSS profiling and phase-sensitive radar measurements) carried out in December 2012, December 2014 and January 2016 (Fig. 1).

The ground-penetrating radar profile shown in Fig. 8a (located in Fig. 1) was acquired in 2016 with a 20 MHz pulsed radar (Matsuoka et al., 2012). The data are geolocated with kinematic GNSS and migrated using Kirchoff-depth migration with a velocity-depth function that accounts for the low firn-air content in this area. More details about acquisition and processing of the radar data are given in Drews et al. (2015). We use the radar ice thickness to validate the hydrostatic ice thickness (Sect. 4.1).

We use three sets of kinematic GNSS profiles that were recorded at 1 Hz intervals with geodetic, multi-channel receivers moving at a speed below 12 km h$^{-1}$. In December 2012, a 20$\times$25 km GNSS network was acquired at the front of the ice shelf (Drews, 2015). The profiles cross ice-shelf channels multiple times. Two years later in December 2014, a 100 km-long North-South GNSS transect was acquired (Lenaerts et al., 2017). The last GNSS dataset was acquired in January 2016, along and across an elliptical surface depression (Sect 3.2). All GNSS elevations are de-tided using the circum-Antarctic tide model

(CATS2008a_opt) from Padman et al. (2002, 2008). Datasets from 2012 and 2016 are processed differentially, relative to a non-moving base station (Drews et al., 2015), while data from 2014 are post-processed with Precise Point Positioning. Elevations from GNSS are used (i) to determine the size of the Gaussian filter applied to the TanDEM-X DEMs (2012 survey, Sect. 2.4), (ii) to assess the accuracy of the TanDEM-X DEMs (2012 and 2014 surveys, Sect. 4.1) and (iii) to extend the time period of surface elevation change detected by the TanDEM-X mission (2016 survey, Sect. 3.2 and 5; Figs. 7 and 8).

BMB was measured at point locations using a phase-sensitive radar. Processing and acquisition schemes are as outlined previously (Nicholls et al., 2015; Marsh et al., 2016). The radar antennas were positioned at 22 sites. Each site was remeasured after 10 days at the same location at the surface (in a Lagrangian framework). This way, relative thickness changes due to strain thinning and basal melting can be detected within millimetres. Strain thinning is corrected using a linear approximation of the vertical strain rate with depth, based on tracking the relative displacement of internal reflectors. The strain correction of the

BMB rates is small ($6.6 \times 10^{-3}$ a$^{-1}$ on average), because strain thinning is small.

## 3   Results

### 3.1   Large-scale pattern of the basal mass balance

The LBMB rates range from -14.7 to 8.6 m a$^{-1}$ (excluding outliers with 0.1 and 0.99 percentiles) and average -0.8 m a$^{-1}$ (negative values signify melting, positive values refreezing). For the 9227 km$^2$ covered by the TanDEM-X DEMs, net mass

loss at the ice-shelf bottom is 6.7 Gt a$^{-1}$. Most melting occurs just seaward of the grounding zone where the western Ragnhild Glacier feeds into the Roi Baudouin Ice Shelf (Fig. 4, label A). This area corresponds to the thickest and fastest part of the

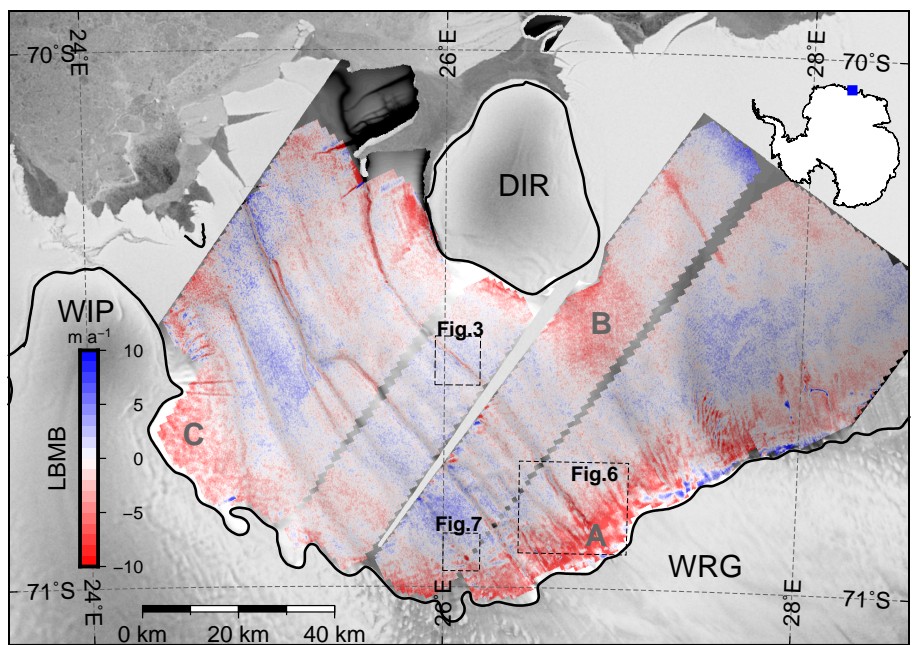

**Figure 4.** Lagrangian Basal Mass Balance (LBMB) of the Roi Baudouin Ice Shelf. Red and blue colours indicate basal melting and refreezing, respectively. The 3 dashed boxes locate the close-ups presented in Figs. 3, 6 and 7. Labels A-C pinpoint areas discussed in the text. Acronyms stand for DIR: Derwael Ice Rise, WIP : Western Ice Promontory and WRG: West Ragnhild Glacier. The LBMB overlays the 2014 TanDEM-X DEM. The background is from the Radarsat mosaic (Jezek and RAMP-Product-Team, 2002) and the black line delineates the grounding line (Depoorter et al., 2013).

grounding zone (Fig. 2e and c). We also find elevated melting close to the western ice promontory (Fig. 4, label C) and on the southern side of Derwael Ice Rise (Fig. 4, label B).

The uncertainties of the absolute LBMB are typically higher than the LBMB itself, because errors unfavourably propagate in mass budgets (Moholdt et al., 2014). Here, we assess a lower bound of the LBMB errors by using the difference fields of
5 the individual LBMB frames in overlapping areas. These show no systematic patterns and the standard deviation amounts to 2.3 m a$^{-1}$. Moreover, comparing the (yearly averaged) LBMB values with the 22 on-site phase-sensitive radar measurements, reveals differences of 1.1$\pm$2.6 m a$^{-1}$ in mean and standard deviation, respectively. We discuss this comparison in more detail in Sect. 3.2. Figures 2b,d,f illustrate the terms entering Eq. (1c), namely surface mass balance, ice-flow divergence and Lagrangian thickness change, whereas Figs. 2a,c,e display the most critical input variables needed to compute those different terms, i.e.,
10 firn-air content, ice velocity and hydrostatic thickness. For the RBIS, the Lagrangian thickness change dominates the BMB (as in Shean et al., 2017), while ice-flow divergence and SMB are both one order of magnitude lower. Qualitatively the large-scale pattern agrees well with the results from Rignot et al. (2013) who also found the highest melt rates close to the grounding line, both for steady state or transient approximations.

To illustrate the advantages of the Lagrangian approach, Fig. 5 shows the Eulerian thickness change, flux divergence and Eulerian BMB. While the large-scale pattern of the Eulerian BMB agrees very well with that of the LBMB, the Eulerian approach fails in the vicinity of ice-shelf channels (arrows in Fig. 5). Advecting topographic features imprint the Eulerian thickness changes (Fig. 5a), however, the Eulerian approach does not fully account for this advection of thickness gradients ($\boldsymbol{u} \cdot \nabla H_i$) in the flux divergence (Fig. 5b). This results in spurious Eulerian BMB in the vicinity of ice-shelf channels (Fig. 5c). These spurious signals in the Eulerian BMB become even stronger when thinning/thickening rates are taken from external datasets which are spatially less well resolved. Using ice-shelf wide, average values (e.g. repeat satellite altimetry) does not account for the advection of ice-shelf channels and other (transient) features in the ice-shelf, hence introducing artifacts in the basal mass balance pattern.

## 3.2  Small-scale variability of the basal mass balance

The larger scale LBMB pattern (>10 km) is overlain by smaller-scale variability. Ice-shelf channels appear most clearly in the DEMs and thus in the hydrostatic thickness fields (arrows in Fig. 2e). In some places, they also co-locate with areas of lateral inflow (i.e., negative flow divergence; arrows in Fig. 2d) and Lagrangian thinning (i.e., negative Lagrangian thickness change in Fig. 2f). In the LBMB, ice-shelf channels appear partially as narrow bands of intense melting. Figure 6 shows one example where ice preferentially melts at the flanks of an ice-shelf channel. LBMB rates drop to -5 m a$^{-1}$ at both flanks, whereas outside the channel the LBMB is close to zero. The slight refreezing found at the channel's apex (1.5 m a$^{-1}$) is very close to the detection limit and its magnitude is 3 times lower than what is observed at the flanks.

Another example of a small-scale feature is illustrated in Figs. 7 and 8. Here, we observe a 0.7×1.3 km elliptical surface depression that is up to 10 m lower than its surroundings and located on the upstream end of an ice-shelf channel. The surface topography also exhibits secondary elongated surface depressions that are shaped like fingers merging into the elliptical depression. We surveyed this area in 2016 with kinematic GNSS profiles, ground-penetrating radar and 22 point-measurements of the BMB with phase-sensitive radar (Sect. 2.7). Lenaerts et al. (2017) identified this feature as one of the 55 features on the Roi Baudouin Ice Shelf, that can be linked to the formation of englacial lakes near the grounding line. They proposed that these features are initially formed as supra-glacial lakes in the grounding zone due to katabatic wind-albedo feedback. Freezing at the lake surface and subsequent burial by snowfalls form at first englacial lakes that are advected farther downstream. As a function of the advection time the liquid water then likely fully refreezes. For the elliptical surface depression considered here, the radar data show a bright reflector at approximately 30 m depth and no coherent signals appear at larger depths (Fig. 8a). We tentatively interpret the bright radar reflector as a refrozen surface of a former supra-glacial lake. The specularity of this interface hinders deeper penetration of the radar signal. However, a more detailed radar analysis is warranted to unambiguously clarify the origin and history of this feature. Here, we restrict ourselves to the elliptical surface depression where we observe significant surface lowering.

The elliptical depression appears prominently in our LBMB field with rates as low as -12 m a$^{-1}$ (Figs. 7b and 8b). On the eastern side of the depression, the BMB from the phase-sensitive radar (Fig. 8b) agrees well with the LBMB estimate, both methods averaging about -0.5 m a$^{-1}$ with little spatial variability. On the western side – which contains the finger-shaped

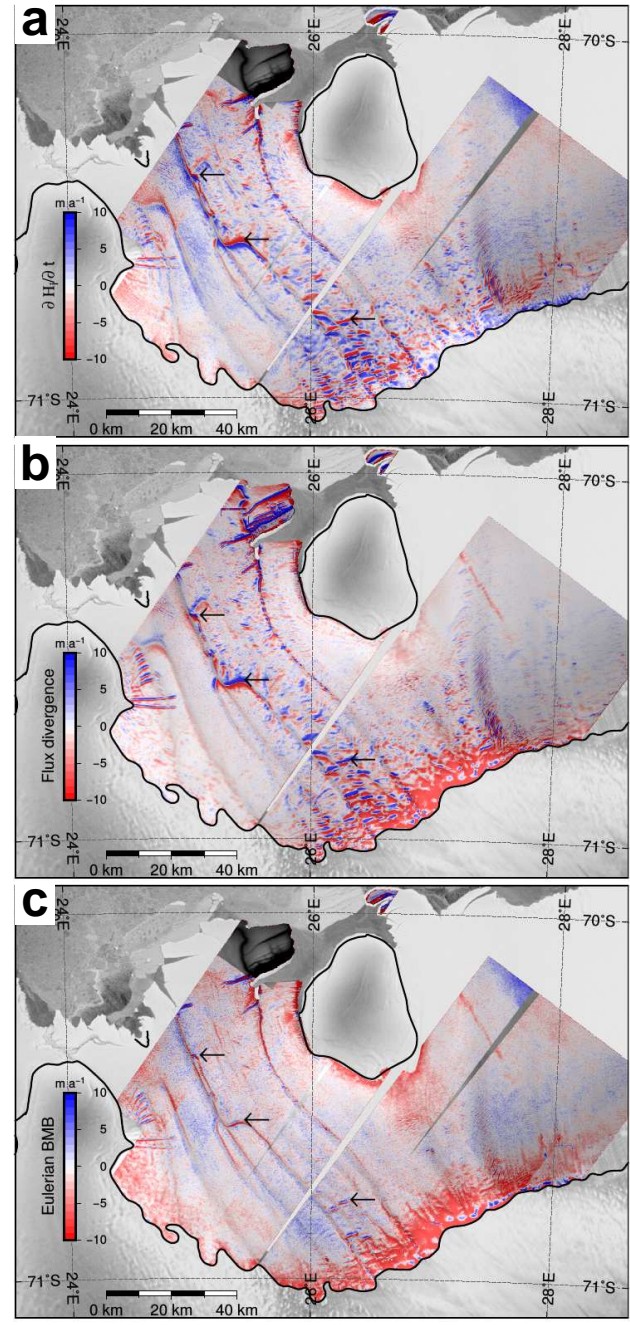

**Figure 5.** (a) Eulerian thickness change ($\partial H_i/\partial t$) (b) Flux divergence ($\nabla \cdot (H_i \boldsymbol{u})$) and (c) Eulerian basal mass balance (BMB). Arrows point to spurious signal due to advection of ice-shelf channels. The background is from the Radarsat mosaic (Jezek and RAMP-Product-Team, 2002) and the black line delineates the grounding line (Depoorter et al., 2013).

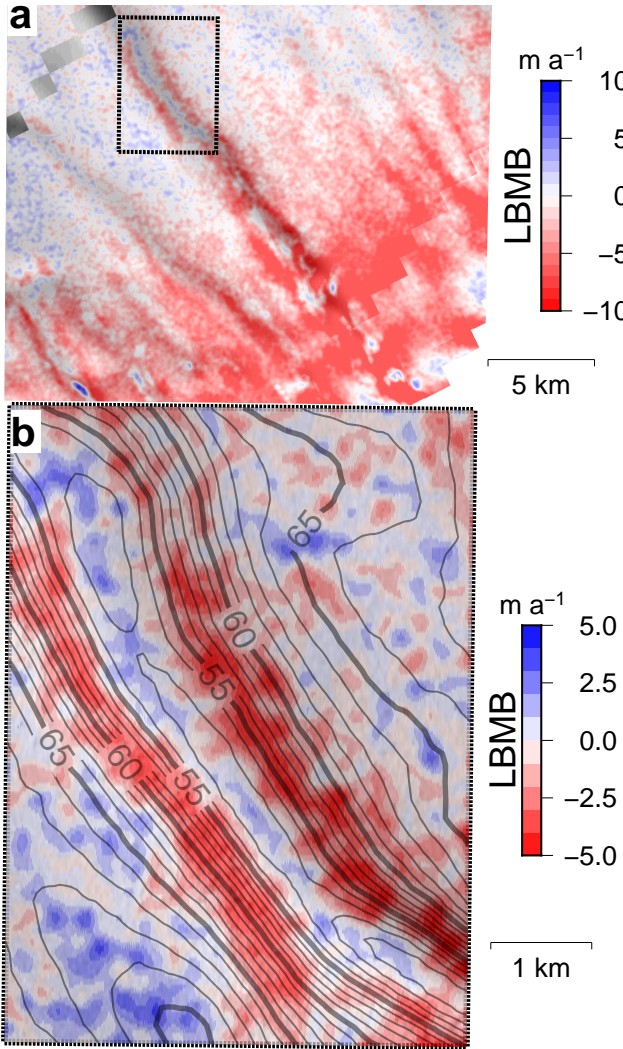

**Figure 6.** (a) Lagrangian basal mass balance around an ice-shelf channel near the grounding line. The box is located in Fig. 4. (b) Close-up view of the box delineated in (a), with 1 m elevation contour lines. Enhanced melting is observed at the channel's flanks.

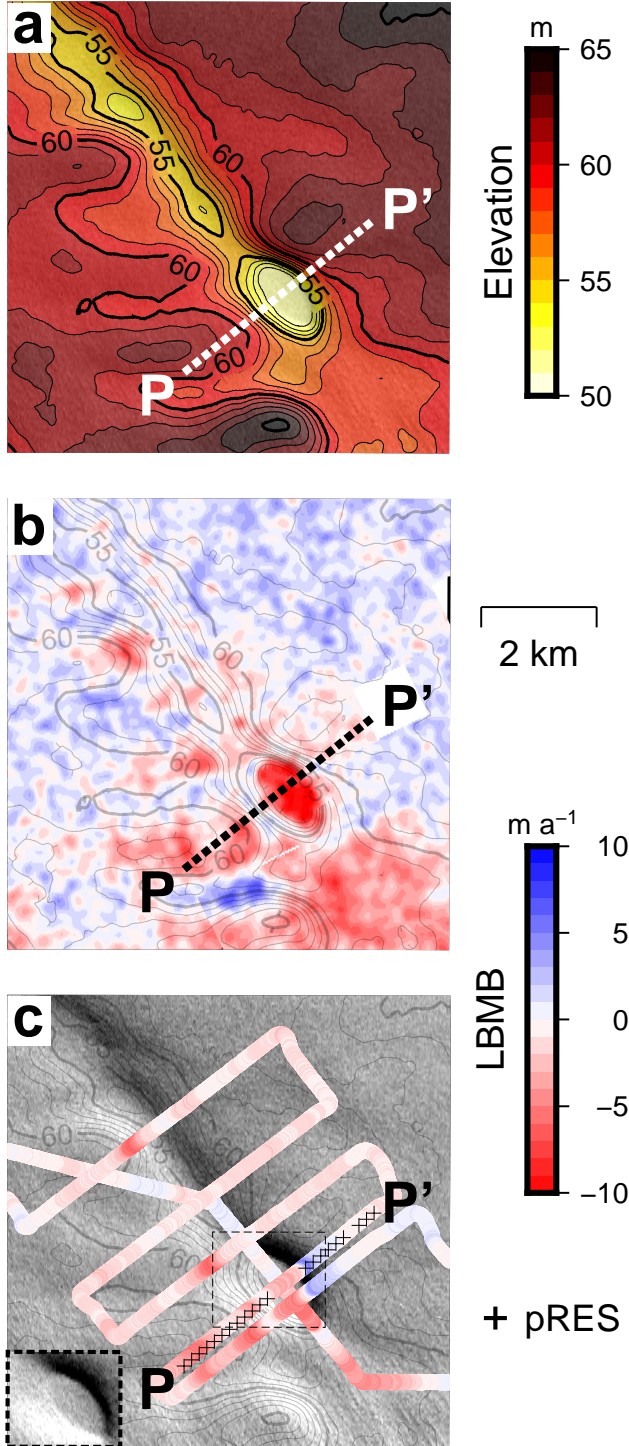

**Figure 7.** Close-up of the elliptical surface depression, located in Fig. 4. (a) Surface elevation from TanDEM-X DEMs from 2014. (b) LBMB (c) Landsat image of 2014 overlaid with the LBMB computed with elevations from the 2014 TanDEM-X DEMs and the 2016 GNSS profiles (using velocities to shift GNSS elevations backward). The crosses locate phase-sensitive radar (pRES) points. The profile PP' is shown in Fig. 8. All subfigures are overlain with the surface elevation contour lines of 1 m.

surface features – larger differences and variability occur. The differences could reflect the more complex topography and/or temporal variations. The large negative LBMB rates in the elliptical depression reflect persistent surface lowering of 0.5 to 1.4 m a$^{-1}$. Ice-flow divergence is negligible at that location. We extend the time series from the TanDEM-X DEMs to 2016 with the GNSS profiles (Fig. 8c) where we find the same localised lowering. This indicates that the high-resolution TanDEM-X DEMs reliably pick up surface elevation changes on sub-kilometre scales. Some of the finger-shaped surface depressions also show surface lowering, but less pronounced than what is seen in the elliptical depression itself. The flanks of the surface depression are significantly steeper on the eastern compared to the western side. Unlike the elliptical depression, the ice-shelf channel located farther downstream does not actively experience melting or refreezing. Away from ice-shelf channels or other surface depressions, our assumptions for the LBMB (such as hydrostatic equilibrium) likely hold explaining the comparatively good fit with the phase-sensitive radar measurements. Inside the elliptical depression, the observed surface lowering cannot unambigously be attributed to basal melting. Regardless of the specific mechanisms causing the surface lowering, this example highlights that much of the small-scale variability seen in the resulting LBMB field can be used to investigate sub-kilometre-scale ice-shelf processes that do not necessarily occur at the ice-shelf base.

## 4 Error sources

### 4.1 Hydrostatic thickness and Lagrangian thickness change

The Lagrangian thickness change is the dominant error source of the LBMB for the Roi Baudouin Ice Shelf, since the magnitude of both ice-flow divergence and SMB are one order of magnitude smaller (Fig. 2). The Lagrangian thickness change depends (i) on factors controlling the hydrostatic ice thickness, i.e. the surface elevation (above sea level), the seawater and ice densities, the depth of the firnpack and temporal variations thereof; and (ii) on the Lagrangian matching of the DEMs following the ice flow. It should also be clear that our approach is only able to detect basal changes reflected in the surface elevations, because ice thickness is derived from hydrostatic equilibrium.

**Calibration and accuracy of TanDEM-X elevations**

The interferometric DEMs provide excellent spatial resolution at the cost that they require calibration. It is straightforward to offset the DEMs to account for the relative phase unwrapping using Antarctic-wide DEMs based on altimetry. More challenging are residual phase trends that may originate from imprecise satellite orbits/SAR processing (Drews et al., 2009) or represent unaccounted tilting of the ice-shelf surface due to tides. In our case, these trends are near-linear and become evident in the difference fields of overlapping DEMs from both different years and from the exact same date and satellite path. In the former, systematic biases extend in the azimuth direction with residual height differences typically ranging from -0.5 to +0.5 m. Such biases strongly imprint the corresponding LBMB fields resulting in a mosaic with linear trends typically ranging from -10 to +10 m a$^{-1}$ in the azimuth direction and differences exceeding 13 m a$^{-1}$ across seams. To account for this, we correct the 2014 DEMs with plane fitting (Sect. 2.4).

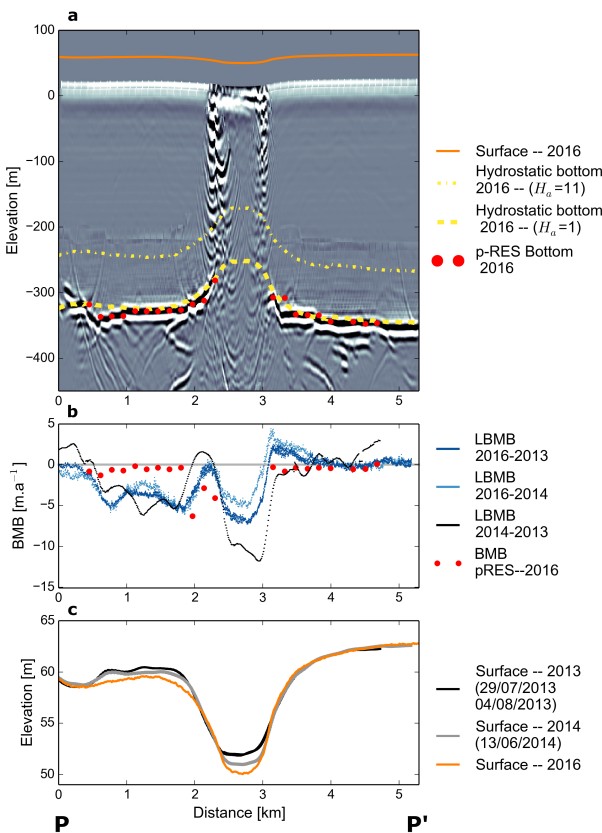

**Figure 8.** Profile PP' across the elliptical surface depression located in Fig. 7. (a) Ice thickness from profiling and phase-sensitive radars together with the hydrostatically inverted surfaces from 2016, measured with GNSS. (b) Different time slices of the basal mass balance. Data from 2013 and 2014 are based on the TanDEM-X DEMs, data from 2016 use GNSS surface elevations. (c) Surface lowering at the elliptical depression: surface elevation between the 2016 GNSS profile and the TanDEM-X profiles from 2013 and 2014. Elevations are referenced to the WGS84 ellipsoid and all profiles are shown in Lagrangian coordinates

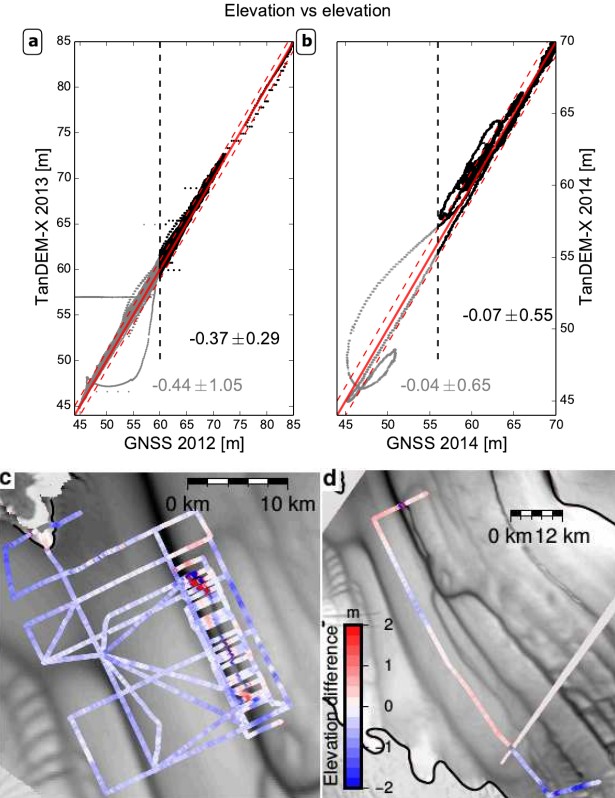

**Figure 9.** (a – b) Comparison between GNSS 2012 – TandDEM-X 2013 and GNSS 2014 – TanDEM-X 2014, respectively. The GNSS data are located in Fig. 1. The grey points, on the left on the vertical dashed lines, lie in ice-shelf channels and are shown in light blue and light red in the profiles in Fig. 1. The plain and dashed red lines show the perfect equality between the two elevations datasets and the ±1 m difference, respectively. (c – d) spatial variations of elevation differences between GNSS 2012 – TandDEM-X 2013 and GNSS 2014 – TanDEM-X 2014, respectively. Background is from TanDEM-X elevations.

We do not correct for systematic trends in individual TanDEM-X frames from the same dates (Fig. S3), not only because the discrepancies are smaller, but also because the small overlapping areas would amplify plane-fitting errors dozens of kilometres away. The standard deviation of the difference fields reduces to 0.3 m after plane fitting. An exception is the two northernmost difference fields, where a trend ranging from -0.8 to 0.8 m remains. In addition to residual phase trends, discrepancies of ~0.5 m
5  can occur in areas where surface slope is locally elevated (e.g ice-shelf channels or surface ridges). Altogether, we therefore estimate the SAR processing uncertainties to be in the order of 0.5 m.

Next, we compare the 2013 and 2014 DEMs with kinematic GNSS profiles from 2012 and 2014, respectively. The time lag between the satellite data acquisition and the collection of ground-truth data is hereby 8-10 months for the 2013 DEMs, and 5-6 months for the 2014 DEMs. For 2012-2013, differences are -0.44±1.05 m, and for 2014 -0.04±0.65 m. The largest
10  discrepancies occur in both datasets near ice-shelf channels were ice advection within the multiple months time lag is significant

(Figs 9). Removing those areas reduces the discrepancies to -0.37±0.29 m in 2012-2013, and -0.07±0.2 m in 2014. Ignoring the dynamic influence of ice-shelf channels, the highest discrepancies are found in the most upstream part of the 2014 GNSS profile (Fig. 9d). There, TanDEM-X elevations are systematically overestimated by up to 2 m near the grounding line. We attribute this bias to decreasing penetration of the TanDEM-X signal, as the firn-air content decreases towards the grounding

zone (Fig. 2a). The X-band radar signal can penetrate up to 8-10 m in cold dry snow (Humbert and Steinhage, 2011), and the bulk part of such a signal penetration would be accounted for during our offset correction. However, errors due to spatial variations of signal penetration remain but affect both the 2013 and 2014 DEMs. To conclude, we estimate that in most areas the relative accuracy of the TanDEM-X DEMs is in the sub-meter range. Errors are slightly elevated in areas where the local surface slope is high, and surface elevation is systematically and significantly overestimated by up to 2 m in a narrow belt close

to the grounding line.

**Hydrostatic inversion**

The main uncertainties for the hydrostatic inversion are referencing the surface elevation to height above sea level, and accounting for density variations. The former depends on the geoid, the mean dynamic topography, tides, atmospheric pressure variations and eustatic sea level. Drews (2015) estimates errors in the geoid and the dynamic topography for RBIS to be within

±1 m. We account for tides and atmospheric pressure variations implicitly by offsetting the TanDEM-X DEMs to the CryoSat-2 DEM, which contains these corrections. The smallest component in the error budget are changes in eustatic sea level rise, which we neglect.

Variations in firn-air content are important because these propagate with a factor of 9 into the hydrostatically inverted ice thickness (Eq. 2). We illustrate this point along profile PP' where the inferred thickness from radar profiling and from

phase-sensitive radar agree closely, but the hydrostatic thickness is >80 m thinner (Fig. 8a). Because surface elevation is well constrained by our kinematic GNSS profiles (Fig. 8c), we attribute this large, unphysical mismatch to an overestimation of the firn-air content. The firn densification model predicts a value of 11 m at that location. However, in the field it became evident that this area is close to a spatially extensive blue-ice area where firn-air content is negligible. Reducing the firn-air content to 1 m reconciles the hydrostatic ice thickness with the observed radar ice thickness (Fig. 8a). Such a large deviation of the

modelled firn-air content may be site-specific because it is located in the transition zone where turbulent mixing by the katabatic winds and a wind-albedo feedback form a micro-climate that causes extensive surface melting with not yet fully understood effects on the firn densification (Lenaerts et al., 2017). The impact of the firn-air-content misestimation on the derivation of the hydrostatic ice thickness is further discussed in Lenaerts et al. (2017). Moreover, Drews et al. (2016) used wide-angle radar measurements in conjunction with ice coring and found that firn density varies spatially over tens of kilometres scales,

in particular across ice-shelf channels, where surface melt water collects in the corresponding surface depressions and locally refreezes. Therefore, we anticipate that at least some of the variability seen in the LBMB field is due to unresolved variations in firn density.

Because of unaccounted variations in firn density, and uncertainties in referencing the freeboard height, our ice thickness field has a lower bound error of at least ±25 m (Drews, 2015). In some areas the error can be considerably larger. However,

the corresponding impact on the inferred LBMB rates is mitigated by the low ice-flow divergence rendering the magnitude of ice thickness less important (Eq. (1c)).

**Lagrangian matching**

Computing the Lagrangian thickness change, requires matching the DEMs to account for ice advection. We use a normalised cross-correlation to match 5×5 km patches from 2013 to the 2014 geometry (Sect. 2.5). Alternatively, the matching can be based on the surface flow field (Moholdt et al., 2014). For the DEMs, this methods yields similar results in terms of the large-scale LBMB pattern, but introduces erroneous positive/negative patterns near ice-shelf channels. This is because the flow velocities are not sufficiently constrained for the flow direction, and tilts by a few degrees cause a significant mismatch in areas where thickness gradients are larger. On the other hand, the 2016 GNSS have to be matched with the velocities, because 2D cross-correlation fails with profiles.

## 4.2 Ice-flow divergence: the benefits of regularized derivatives

The high-resolution velocity field is too noisy in magnitude to approximate the derivatives in the flow divergence with finite differencing of neighbouring cells (gridded to 125 m posting). This can be accounted for by smoothing the velocity field prior to taking the derivative. However, this type of smoothing can blur abrupt changes in the flow velocities and corresponding strain rates. This is important, because we suspect that ice-flow velocities change abruptly in ice-shelf channels that experience strong basal melting (Drews, 2015). We, therefore, explore the use of total-variation regularization which treats abrupt (and discontinuous) changes more accurately (Chartrand, 2011). Figure 3 illustrates a close-up of an ice-shelf channel (inset "Fig.3" in Fig. 4) where we compare the "Normal" (unsmoothed) velocity divergence (b) with its regularized (c) and smoothed (c-e) versions. For the latter, we applied average filters of 375×375 m, 1,125×1,125 m and 1,875×1,875 m (i.e. kernels of $3 \times 3$, $9 \times 9$ and $15 \times 15$ pixels, respectively) to the velocity field, before computing the gradients. The enhanced velocity-divergence has a similar magnitude in the regularized and the smoothed version using a $375 \times 375$ m window. However, the latter is noisier outside the ice-shelf channel than the regularized version. In the regularized case, velocity divergence at the channel's apex is 8%, 24% and 40% lower than for the 375×375 m, 1,125×1,125 m and 1,875×1,875 m kernels, respectively. However, the inferred LBMB rates are insensitive to the technical implementation of the derivatives, because the Lagrangrian thickness change controls the signal at RBIS. Nevertheless, in order to study the dynamics of the smaller-scale ice-shelf channels, efficiently denoising the derivatives becomes increasingly important, in particular for ice shelves where the dynamic thinning terms is more important.

## 4.3 Surface mass balance

Both the firn-air content and the SMB are spatially less well resolved than our ice thickness and velocity fields. Consequently, we do not capture their spatial (and temporal) variations on the length scales associated with ice-shelf channels. Both Drews et al. (2016) and Langley et al. (2014) found evidence in the shallow radar stratigraphy that the SMB may be locally elevated

in those areas, potentially reflecting the deposition of drifting snow at the bottom of surface slopes (Frezzotti et al., 2007). If this holds true, then the systematic underestimation of the SMB would result in a positive bias of the LBMB in those areas.

## 5    Discussion

The large-scale patches of enhanced basal melting (Sect. 3.1; labels A-C in Fig. 4) are sufficiently far away from the tidal bending zone so that we can safely assume hydrostatic equilibrium. These regions are also detected by Rignot et al. (2013), based on different input datasets (i.e. Eulerian thickness change based on ICESat-1). Patches A-C line up with deepest parts of the ice-shelf base and the largest gradients in the hydrostatic ice thickness. A large ice draft fosters basal melting because the freezing point is lower with depth (e.g. Holland et al., 2008). The steep basal slopes facilitate entrainment of heat in the mixed layer beneath the ice shelf increasing basal melting (Jenkins and Doake, 1991; Little et al., 2009).

The smaller-scale variations in LBMB are more difficult to interpret, because these are overlain by unaccounted variations in firn density, SMB, and ice that is not in hydrostatic equilibrium. Nevertheless, the comparison with the phase-sensitive radar data and the kinematic GNSS profiling increases our confidence that much of the relative variability that we observe here is meaningful. The surface lowering of the elliptical surface depression is consistently observed over a 3-year time period marking this zone as dynamically active. Two other options are: (i) a transient adjustment of the surface towards hydrostatic equilibrium (Humbert et al., 2015) as a response to some unknown event in the past which locally reduced the ice thickness, and (ii) the surface lowering may reflect vertical creeping of a liquid water body through the ice column. In any case, the surface lowering is restricted to a small area and the ice-shelf channel farther downstream appears passive (i.e. does not show significant melting nor refreezing).

In most areas, ice-shelf channels at RBIS seem to advect passively and basal melt rates there do not significantly stand out from those in the larger surrounding. Exceptions are the locally elevated basal melt rates in ice-shelf channels in the interior of the RBIS (e.g. inset "Fig. 3" in Fig. 4) and close to the grounding zone (Fig.6 and its corresponding inset in Fig. 4). Almost all ice-shelf channels at RBIS are connected to the grounding line and may arise from water-filled subglacial conduits injecting subglacial-melt water into the ice-shelf cavity, driving a spatially localised buoyant melt-water plume (Jenkins, 2011; Le Brocq et al., 2013; Drews et al., 2017; Sergienko, 2013). Such localised melting near the grounding zone has been previously observed on Pine Island Ice Shelf using similar methods as done here (Dutrieux et al., 2013). However, on Pine Island Ice Shelf, background melt rates are an order of magnitude larger than what is observed here (Depoorter et al., 2013; Rignot et al., 2013) and Dutrieux et al. (2013) analysed DEMs separated by 3 years (compared to the 1 year time period used here). This explains why locally elevated BMB values appear more clearly on other ice shelves. We find some evidence that basal melting is concentrated on the flanks, rather than on the apex (Fig. 6). This accords both with observations (Dutrieux et al., 2014) and modelling (Millgate et al., 2013). Dutrieux et al. (2014) suggest that the presence of a colder water blocks the heat flux from below near the apex of the channel. Alternatively, modelling suggests (Millgate et al., 2013) that a geostrophic current develops beneath the channels (if the channels are wide enough) which preferentially melts at the channel's flanks. This seems less likely here because ice-shelf channels near the grounding line are narrow (i.e. a few hundred meters wide and high).

In summary, our observations suggest that the LBMB varies on multiple spatial scales which has several implications. First, point measurements with phase-sensitive radars are not necessarily representative for a larger area. Particularly in areas where thickness gradients are large, phase-sensitive radar measurements are best understood in combination with satellite-based estimates covering larger spatial scales. On the other hand, on-site point measurements are crucial to estimate the quality of

the satellite-based BMB estimates, which are uncertain in their magnitude. Second, this sub-kilometre variability in ice-ocean processes poses challenges for coupling ice flow with ocean models, because highly resolved ocean models and community efforts, such as the Marine Ice Sheet–Ocean Model Intercomparison Project (MISOMIP), are typically gridded with 1-2 km (Dinniman et al., 2016; Asay-Davis et al., 2016). This is too coarse to capture the spatial variability that we observe here.

## 6    Conclusions

We derived the Lagrangian Basal Mass Balance (LBMB) of the Roi Baudouin Ice Shelf by combining TanDEM-X DEMs of 2013 and 2014 with high-resolution surface velocities and atmospheric modelling outputs. On a large scale, the LBMB shows the highest basal melt rates where the ice draft is deepest and steepest, i.e. close to the grounding line and near Derwael Ice Rise and the Western Ice Promontory. This pattern is overlain with significant sub-kilometre scale variability, as witnessed by localised surface lowering of an elliptical surface depression and large basal melting rates below some sections of ice-shelf

channels. For the latter, we find evidence that at least in some areas, basal melting is concentrated on the channel's flanks as opposed to its apex. Key advancements in our methodology to elucidate this variability are (i) the calibration of the DEMs to account for residual trends from the interferometric processing, (ii) the quality of the matching procedure – using normalised cross-correlation coefficients – for calculating the Lagrangian thickness change , and (iii) the total-variation regularization of the spatial derivatives that preserves abrupt changes in flow velocities that are sometimes observed across ice-shelf channels.

New satellites (such as TanDEM-X or Sentinel 1) will continue to provide highly-resolved datasets of surface elevation and ice velocity. In comparison, atmospheric modelling does not (yet) provide the required spatial resolution on firn-density and SMB to solve the mass budget reliably on sub-kilometre scales. Although the uncertainty of the absolute LBMB values remains high, we find a good fit with on-site measurements from phase-sensitive radar, and we demonstrate that much of the spatial LBMB variability contains information about ice-shelf processes occurring at sub-kilometre scales. This variability highlights the

complexity of the ice-ocean and ice-atmosphere interactions on small spatial scales on ice shelves, which need to be accounted for by glaciologists, oceanographers and atmospheric scientists.

*Acknowledgements.* This paper forms a contribution to the Belgian Research Programme on the Antarctic (Belgian Federal Science Policy Office), project SD/CA/06A (Constraining Ice Mass Change in Antarctica, IceCon). S. Berger is supported by a FRS-FNRS (Fonds de la Recherche Scientifique) "Aspirant" PhD fellowship. R. Drews was partially supported by the Deutsche Forschungsgemeinschaft (DFG) in the

framework of the priority programme "Antarctic Research with comparative investigations in Arctic ice areas" by the grant MA 3347/10-1. S. Sun is supported by the FNRS-PDR (Fonds de la Recherche Scientifique) project MEDRISM. TanDEM-X data originate from German Aerospace Center (ATI-GLAC0267). We thank Nicolas Bergeot (Royal Observatory Belgium) who helped with GNSS processing and K.

Nicholls for his valuable help in processing of the phase-sensitive radar. We received excellent logistic support by the Belgian Military, AntarctiQ and the International Polar Foundation during the field campaigns. Finally, we thank J. Lenaerts and S. Ligtenberg for sharing results from atmospheric modelling, as well as D. Shean and G. Moholdt for their constructive comments on this manuscript.

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
