# Peer review of "Detecting high spatial variability of ice-shelf basal mass balance (Roi Baudouin ice shelf, Antarctica)"

_The Cryosphere, 2017_

## Referee Comment (RC1)

**Detecting high spatial variability of ice-shelf basal mass balance (Roi Baudouin ice shelf, Antarctica)**
S. Berger et al
TCD 2017
doi:10.5194/tc-2017-41

**Summary**

This paper documents the high-resolution basal mass balance for the Roi Baudouin ice shelf for 2013-2014 using TanDEM-X DEMs.  The authors provide a good description of the Lagrangian basal mass balance derivation (building on previous literature for analyses of other ice shelves), and compare DEM-derived basal mass balance with in situ pRES data and GPS-derived basal mass balance.  In general, the methodology, results, and figures are well done.  There are several issues that require further attention before publication.  The authors claim to use a novel method to improve quality of velocity divergence products.  The resulting products still contain fairly significant noise and the authors provide only one comparison with a coarse smoothing approach which generally looks similar.  There are some significant discrepancies between firn air content from a dynamic firn densification model (RACMO-FDM) and the in situ radar data, with different assessments of significance.  The authors offer a detailed analysis of an ice-shelf surface depression, which they interpret as a moving, unfrozen englacial lake 30 m below the surface.  There is limited evidence to support this speculative interpretation.  The LBMB derived from pRES/GPS measurements shows some significant differences when compared to the DEM-derived LBMB.  On the whole, this is a good paper that should be published after these and other issues are addressed.

**General comments**

The abstract ends with a mention of challenges for full coupling between ice and ocean models, but this is never addressed in the paper.

Methods section needs some reorganization.  Recommend separating DEM generation from hydrostatic ice thickness calculation.

No meaningful analysis of TDM DEM accuracy is provided, only "estimated relative vertical accuracy better than 1 m (based on the standard deviation in overlapping areas)" – where are these areas?  Are they all flat, or does this include higher surface slopes (like the surface depressions over channels discussed in the text)?  Are you certain these areas are not evolving during the 2013-2014 period due to SMB or dynamics?  A much more convincing approach would be to show a figure of DEM standard deviation over static surfaces (exposed bedrock, slow-moving ice with limited SMB).  No dates (month, day) are provided for the TDM DEMs, and we don't know over what period the "32 from 2013" and "11 from 2014" actually cover.  Were mosaics generated for each year, centered on Jan 1 of the respective year?  Did you compute Lagrangian Dh/Dt for each pair of DEMs, or for the annual mosaics?  What about offsets between DEMs within a single year – if they are months apart, won't there be significant advection for areas flowing 300 m/yr?

Are you convinced that you are not seeing penetration of the radar into snow and ice in the TandDEM-X DEMs? Other studies have shown this can be several meters, potentially up to 10 meters in cold, dry snow. This could impact the comparisons with GNSS surface elevations.

The authors indicate that they smoothed their input 10-m posting TanDEM-X DEMs with a gaussian filter (no filter dimensions provided, but 7-sigma implies a large kernel). This inherently reduces the resolution of these DEMs, and will smooth edges of small-scale features like channels.

It is important to be clear that the DEMs are measuring surface elevation and the surface expression of features interpreted as basal channels. You are not directly measuring channel depth using surface elevation.

Are there any airborne radar profiles (OIB, BAS, etc.) over this shelf? Seems like there must be something available. If so, you can use observed ice thickness and deviation from floatation thickness calculated from your surface elevations to estimate firn air content [*Holland et al.*, 2011].

The ordering of the figures is a bit odd – the comparison showing divergence for regularization vs. smoothed velocities should be shown after describing the method, or early in the results. The LBMB figure (the main result) should come after the component figure

**Specific comments (and additional general comments that came up for specific sections)**

Page 1:
Line 12: Is the lake 30 m or the surface depression is 30 m below surrounding surfaces?
Lines 16-17:
Line 21: Not sure "Marine Ice Sheet Instability" should be capitalized,

Page 2:
Line 10: delete "the" before "the BMB"
Line 11: "which form"
Line 11-12: I don't understand this 50% claim, also, the sentence ends abruptly – missing "surrounding ice"?
Line 10: What about GPS receivers [*Jenkins et al.*, 2006; *Shean et al.*, 2017]?
Line 14: Channel carving does not cause increased crevassing – these are separate processes
Line 16: delete "at 10 m gridding" – not relevant for intro
Line 17 and later: I believe TC should be "Figure" not "Fig."
Line 22: delete "all the way"
Line 23: delete "In the following,"
Line 24: delete "results", also suggest "with focus on…" rather than "special focus"
Line 26-28: "accounts for" is a bit awkward – you are mitigating the noise, not accounting for it
Line 29: change "observational evidence" to "observations"
Line 29: suggest simplifying to "…validated with phase-sensitive radar, ground-penetrating radar, and GNSS observations."

Page 3:
Equation 1: This is 3 equations! Should probably split. You might start with the standard mass conservation rather than rearranging at the start.
Line 6: "vertically integrated" (no hyphen needed) – perhaps "column-average" is a better descriptor here. You are assuming surface velocity is equal to column-average velocity, this should be stated somewhere
Line 6: "In principle, "

Line 1: Not sure what you mean here - Eulerian dH/dt involves two thickness measurements
Line 6: "can only be adequately done" is subjective
Line 7: Add [*Shean*, 2016; *Shean et al.*, 2017] for additional derivation and discussion of Eul/Lag elevation change
Line 10: "from 2013 and 2014"
Line 10: Suggest deleting ", clearly resolving ice-shelf channels" – you are claiming that the 10-m resolution is novel (a claim some might reject). This resolution can resolve many small-scale features on ice shelves, not just channels.
Why so many details in paragraph 2? Details like "we calculate Lag thickness change by cross-correlating the TanDEM-X DEMs (using 5x5 km2 patches…) should be in Seciton 2.5.
5x5 km$^2$ – I think you mean 5 km x 5 km.
OK, so you have 10-m DEMs, and you are cross-correlating using a 500x500 pixel kernel. Using a kernel this large inherently reduces the resolution of the output velocity products. Did you do this for every pixel, or for some sparse interval?
Line 17: "freely floating"
"viscous inflow in ice-shelf channels" is a process, not a "small-scale feature"
Line 24: "of 1996" and "of 2010" should be "from 1996"
Line 24-25: So, you compared the 1996, 2010 and 2014 velocities and found no evidence for temporal variations? What are dates of the input InSAR? What was time period of GNSS? Surely the velocities aren't identical.
Line 25: "This dataset" – ambiguous
Line 27: Replace "cutting edges" with "seams"
"Offsets between the two datasets are over 60 ma-1 in places."
You are blending seams with feathering, but you are not actually correcting the offsets in velocity magnitude. This will lead to smooth gradients for the divergence, but will likely lead to incorrect horizontal path determination for your Dh/Dt. That 60 ma-1 is a significant portion of the ~50-300 ma-1 velocities over the ice shelf. Your Lag Dh/Dt obs could be "off" by 6 pixels at your DEM resolution.
Line 30: "The SMB is based on…" is awkward start. You are using RACMO, not basing your SMB on RACMO.
Line 32: Need a reference for these processes that RACMO is reproducing. Also, RACMO doesn't predict anything. Add reference to Figure 2b when discussing SMB spatial distribution.

Seciton 2.4: This section should be split into 1) DEM processing and correction and 2) Hydrostatic thickness calculation.
Switching between "were" and "are" – should all be past tense

Add reference to TanDEM-X mission.

Add reference for SARscape software?

How did you co-register the DEMs? How do we know there aren't horizontal and vertical offsets between your input DEM data that will lead to artificial elevation change (and LBMB) signals?

Looks like a seam artifact is present in Figure 1 between labels 3 and B. LBMB values are positive (~3-5 m/yr) and then immediately adjacent, close to 0.

Line 12: Don't start sentence with acronym

Line 11: Based on my experience with Greenland data, I am skeptical of this 1 m vertical accuracy for TDM. This needs stronger justification.

Line 18: What are dimensions of your Gaussian filter – 7 sigma is large, implying a large kernel, which will significantly reduce the resolving power of the output DEM (definitely not 10-m)

Weren't the DEMs and GNSS data collected during different time periods? Are you assuming that no change occurred between the two collections? What is the "mean and standard deviation" of the differences? The GNSS data are isolated to a small area, how do we know that this is representative of the larger ice shelf?

Line 22: Wording is awkward, suggest something like "We calculate freeboard ice thickness assuming hydrostatic equilibrium…"

Line 25: Firn air content accounts for total air content of the firn, not variable firn density.

What are typical mean dynamic topography offsets for this location?

Are you accounting for density errors in your uncertainty estimates? I've found that including a +/- 5 kg/m3 uncertainty in density can dominate the freeboard thickness error, much more than a few meters of firn air content error.

I've found that IMAU-FDM estimates over ice shelves in West Antarctica are biased high, in some places 5-10 m too high, compared to radar-derived firn air content using techniques from

Review was interrupted for several weeks at this point. Picking up again. I apologize for discontinuity in comments.

Line 28: What is approximate mean dynamic topography correction for this location?

Line 5: I think you mean "these approaches are not well-suited…"

Line 9: What is meant by "wiggliness" of the derivative? Need a little more explanation about what you are solving for and how the process works. This is emphasized as a novel method, so it should be documented clearly. So, you are computing horizontal and vertical gradients separately, then combining in a large inversion?

How does the velocity map resolution impact alpha?

When I look at Figure 2D, I still see plenty of noise in the velocity divergence map.

Line 18: replace "such as" with "including"

OK, so you de-tided the GPS surface elevation data. Did you also detide the TDX DEMs?

What depths were the reflectors used to determine strain thinning? Did you have to account for firn compaction?

Page 8:
Lines 1-2. I don't understand this. You are saying the strain correction is small compared to the basal melt rate, with both provided in units of m/yr. This makes sense. The 10-day interval should be irrelevant here – why is it mentioned?
This result about strain correction suggests that the velocity divergence term (and the regularization) is not necessarily important for the larger shelf LBMB calculation. The
Line 6: "seaward"
In Figure 1, what is the band of large positive (blue) values along the grounding line to the right of Label "1"? Artifacts or real refreezing signal?
My guess is that the Depoorter grounding line is in the wrong place, the ice at this location is grounded, and this area should be masked.
Line 8: "stoss" is a relative term – could be leeward for ocean circulation, different direction for wind, different direction for ice flow – suggest changing to absolute direction ("south")
Line 11: Where are these overlapping areas? Are you sure that the LBMB is not changing over the period for which you are performing the analysis (could some of your observed std be due to real changes in melt rates?) I think you are saying that formal error estimates are larger than the magnitude of the measured signal.

The last sentence on Page 8 and first sentence on Page 9 have no real context. I think you are making an argument that dh/dt from sparse or low-res measurements is problematic. But you have high-res DEMs, so you don't need external datasets for high-res Eulerian elevation change.

Line 7: Is this order of magnitude difference present everywhere? Seems like DH/Dt values are close to 0 in the middle of the shelf, so the vdiv and smb terms become much more important.
Line 11: Is this convergence within channels present across the full channel width, or just on the sides, or do you lack the resolution to determine this?

I don't see negative Dh/Dt across all channels in Figure 2f. In fact, I see positive Dh/Dt in several places (e.g., just northwest of the right-most arrow in figure 2d). So, the convergence is causing thickening of the ice shelf at this location? More likely is that you are picking up surface elevation change due to snow redistribution

Line 3: At the channel center, LBMB values are positive, potentially even +2-3 m/yr. This is not close to zero. What is going on here? Is that refreezing (unlikely) or is this an artifact? Need to address this if you are going to interpret the signals on the sides with confidence.
Line 4: not km2 here, just km
Line 4: elliptical, not ellipsoidal
Line 5: You don't necessarily know that the lake is connected. It's location is adjacent to the channel, but careful about wording that could be misinterpreted here.
Line 6-7: This interpretation about tributaries needs more support if it is going to be included.

I haven't seen the Lenaerts et al (2017) paper, but I'm puzzled by this interpretation. Are you suggesting that the lake is liquid water, 30 m below the surface of the ice shelf? Why have these not refrozen?

Line 12: "blocks the penetration" suggest "attenuates"

This interpretation is inconsistent with the radar data shown in Figure 6. There are many reflections beneath the "lake" feature. This to me suggests there is no way that this is liquid water. There may be an interface that is attenuating the radar signal, but definitely not salty water, which seems like the only way to prevent refreezing.

Line 15: I disagree – this interpretation is important. I would not call it an englacial lake if you have no direct evidence for this interpretation. Stick with "elliptical surface depression" and be consistent throughout.

I am not entirely convinced that the apparent LBMB over this feature is not due to variable snow accumulation and redistribution over the periods when you have elevation measurements.

I disagree with the interpretation that the pRES and DEM-derived LBMB values agree "well" or show a "near-perfect" fit. Figure 6b shows major disagreement (+/-2-6 m/yr) between the two in places. These offsets are large and significant compared to the magnitude of the LBMB signal.

This paragraph is very long, and should be broken up

Line 19: I don't think "low" is the right term here, try "large negative"

In Fig 6c, I don't understand why the surface is getting lower over the depression, but getting higher between 0.5-2.0 km. So, if I understand correctly, the P-P' profile was extracted in a fixed Eulerian 2016 location? So some/all of the observed elevation change for this fixed profile could be due to advection? Why was not extract the profile in a Lagrangian sense, moving with the feature?

Are you convinced that you are not seeing penetration of the radar into snow and ice in the TandDEM-X DEMs? Other studies have shown this can be several meters, potentially up to 10 meters in cold, dry snow. This could impact the comparisons with GNSS surface elevations.

It looks like the large negative values in LBMB along the channels is mostly coming from the velocity divergence term. Are you confident that these negative values on channel sides are not artifacts velocity resolution and regularization approach are a

Line 25: Again, no evidence for connection. Inactive in what sense? You don't see an elevation change signal, so it is not actively experiencing melting or refreezing.

Line 28: How does an englacial lake creep through an ice column? Is there a reference for this, or is this your interpretation? Still don't understand how this could remain unfrozen.
Lines29-30: Not sure what this sentence contributes. I'm still thinking that much of the observed elevation change within the depression could be due to local surface accumulation and wind redistribution.

Page 13:
Line 5: This is also highly dependent on the density ratio of ice and ocean water, not just the firn air correction.
Line 11: Do you mean overlapping DEMs?
Line 13: "cutting edges" – I think you mean "differences exceeding 13 m/yr across seams"
Line 15: OK, but isn't there a 2-year time difference between the DEM timestamps and the GNSS measurements?  Is this an appropriate comparison?

These offsets in the firn air content are significant.  What percentage of the total ice thickness is this?  In some cases ~5-10%?

Line 29: Also, can have surface and basal crevasses, filled with ice/water/air that will affect the air content.  Maybe less relevant at these locations.
Line 34: "To get our LBMB in Lagrangian geometry" – I think you mean "to determine the relative offsets between surface features in the two DEMs, needed to compute Lagrangian Dh/Dt…"
These are length and width in km, Not km^2.
Line 35: Correlating DEMs is not always better, esp for smooth, featureless surfaces or sparse altimetry data.

Careful about comparing magnitude of "SMB" here, which is m w.e. – you are using the expected elevation change output by a firn model driven by SMB, right?  Not the same thing.

Line 3: Bimodal is not the right word.  I think you mean a positive/negative signal due to misalignment.
Line 4: So which did you use again for the paper?  I thought you were using existing flow fields, not your cross-correlated flow fields?  Make sure this is clear wherever it is discussed in the text.

The differences between the two approaches in Fig 7 are not as drastic as one would expect.  Why did you choose 5.125 km window?  This is a 41 pixel window for 125 m velocity maps!  I would chose something far smaller, like a 3x3 px, 5x5 or even 9x9 px.  My guess is that such a filter would remove noise but preserve the same channel-scale divergence as the regularization.  It's not really a fair comparison to say that your method is better than smoothing when you only tested one very large smoothing window.  I'd also like to see the original data, so we can assess the improvement offered by the different approaches.

Line 15: OK, earlier you made the argument that the velocity divergence term was small, so the error in firn air and hydrostatic thickness was negligible.  Now you are making the argument that the velocity divergence term is significant.  What is the error from a 10 m error in firn air correction for the entire shelf?  Might be an informative map, which could be used to produce a map of uncertainty in LBMB.  I think Moholdt had some nice figures like this in his paper.
Line 23-24: You're not really applying an atmospheric model, you are using output elevation change products from a firn densification model (FDM) driven by regional climate model (RACMO) outputs.
Line 24: Could also be overestimation, which would have negative bias.

It's important to separate SMB from elevation change due to SMB.  SMB is in m w.e., while the actual elevation change will depend on density of snow and firn.  Fresh snow will have a lower density, and potentially a greater impact on elevation change.

Line 28: What is the approximate length scale of the tidal flexure here?

Line 29-30: Suggest that you state the datasets used – ICESat-1 was used to infer ice thickness.

While slope may very well be an important factor here, the shelf draft is significantly deeper near the grounding line, where we would expect a suppressed freezing point and enhanced melting.  Like some combination of these two factors.

Line 3: "observed variability" in what? LBMB?

Line 4: Why only surface lowering?  Really it's more general – surface elevation that is not representative of hydrostatic ice thickness.

Line 8: "This indicates" – the datasets don't' make the region active

Line 10-11:  I still don't understand how a liquid body can "creep" through the ice column unless it has high salinity.

Line 12: "appears passive" – not sure what you mean by this.  I think you mean that the apparent basal melt rates are small.

Line 15: "connected to the grounding line" – the channels appear to originate near the grounding line, but this does not necessarily imply a direct connection.  Some of the channels can also be inherited from bed topography at the grounding line, which leads to feedbacks in basal melt magnitude/distribution.

Line 17: Be sure to specify that you are talking about subglacial meltwater originating upstream beneath grounded ice.

Page 16:

Line 1: Cite Shean (2016)

[*Dutrieux et al.*, 2013] noted that melting appeared to be focused in channels near the grounding line and on keels near the outer shelf.  The full-shelf thickness gradient at PIG is substantial, 1-1.5 km near the grounding line, and 300-500 m near the calving front.

I believe there is also some component of Coriolis that can lead to asymmetric melt within the channels.

Line 11: "sub-kilometer"

Line 14-15: So, over what length scales are pRES point measurements representative?

Line 16: "uncertain in their magnitude" – so you are suggesting that satellite LBMB on its own is not useful?  I disagree.  I think it's also a matter of LBMB signal magnitude.  At PIG, melt rates exceed 200 m/yr in places, so 5-10 m/yr error is negligible.

Line 18: "we derived"

Line 19: This makes it sound like you ran an atmospheric mode.  Replace "atmospheric modelling" with "elevation change output from a dynamic firn model driven by regional climate model output"

Line 20: deepest and steepest.

Line 22: Really, you are not observing large basal melt rates below ice-shelf channels, you are observing high melt rates beneath surface depressions that form over basal channels

Line 25: I'm still not clear on the matching procedure – did you combine your independent velocities and your velocity maps derived from DEM correlation? Restate the actual procedure that offers improved quality here.

Line 27: "…small-scale flow anomalies (e.g., channel margins)"

Tables:
Table 1: Mixing km and degrees in the "gridding" column

Figures:
I might reorder these figures to build to the LBMB map. You could show the components (Fig 2) and the Eulerian map (Fig 3) first, then LBMB map (Fig 1).

Figure 1: Delete "in slight transparency"

Figure 2: Which surface velocities are you showing – the InSAR-derived products, or your velocities from cross-correlation of TDM?

In panel d, what is the ~30 km long linear feature to the east of DIR? Is this an artifact? There doesn't appear to be any channel in the ice thickness map. I'm guessing this is where you used the Rignot et al velocities. Should add something about this in caption.

Figure 3: What time period is shown here (ie what is dt)? Is this eulerian dh/dt from your 2013 to 2014 DEMs? I don't think you mean steady state here – you are showing observations, right? The shelf could still be thinning/thickening in Eulerian frame.

Figure 5: Mention arrows in caption. Although I don't think these features are necessarily worth noting (there is also a similar depression on the other side of the channel). Are the 5-10 m/yr freeze-on signals (near P) in panel B real?

Figure 6: Why is there a large offset between your GNSS surface (~60 m on the y-axis) and the reflection from the surface in the GPR profile (~10 m on the y-axis)? Shouldn't these be in the same place?

When the GPR data were processed, did you use 10 m or 1 m of firn to convert two-way traveltime to depth?

What dates (month, day) from 2013 and 2014 are you showing here? Don't you have many DEMs from each year over this location?

Figure 7: It would be useful to see another column here with the divergence and LBMB from the original velocity data for comparison, so we can see the improvement offered by your regularization.

I don't understand why there is still so much noise in panel d. This suggests that the elevation change measurements from the TanDEM-X products are the source of most of the noise in the final LBMB maps. Eyeballing this figure, if I were to draw a window around the lower quadrant and compute a standard deviation of these values, it would probably be something like 1-1.5 m/yr. Is this consistent with your stated vertical accuracy of <1 m?

Dutrieux, P., D. G. Vaughan, H. F. J. Corr, A. Jenkins, P. R. Holland, I. Joughin, and A. H. Fleming (2013), Pine Island glacier ice shelf melt distributed at kilometre scales, *The Cryosphere*, *7*(5), 1543–1555, doi:10.5194/tc-7-1543-2013.

Holland, P. R., H. F. J. Corr, H. D. Pritchard, D. G. Vaughan, R. J. Arthern, A. Jenkins, and M. Tedesco (2011), The air content of Larsen Ice Shelf, *Geophys. Res. Lett.*, *38*(10), n/a-n/a, doi:10.1029/2011GL047245.

Jenkins, A., H. F. Corr, K. W. Nicholls, C. L. Stewart, and C. S. Doake (2006), Interactions between ice and ocean observed with phase-sensitive radar near an Antarctic ice-shelf grounding line, *J. Glaciol.*, *52*(178), 325–346.

Shean, D. (2016), Quantifying ice-shelf basal melt and ice-stream dynamics using high-resolution DEM and GPS time series, Ph.D. Thesis, University of Washington, Seattle, WA, 14 July.

Shean, D. E., K. Christianson, K. M. Larson, S. R. M. Ligtenberg, I. R. Joughin, B. E. Smith, and C. M. Stevens (2017), In-situ GPS measurements of surface mass balance, firn compaction, and basal melt rates for the Pine Island Glacer Ice Shelf, Antarctica, *The Cryosphere*, *submitted*.

---

## Short Comment (SC1) · 8 Apr 2017

Informal comments from Rupert Gladstone

Interesting work!

Fig 4, really interesting to see the detail here. Please add a label other than 60 so we don't have to count contours! You might want to add a subplot here showing actual thickness if you have it, or maybe hydrostatic thickness if you don't as the basal expression of this feature is presumably much deeper than the surface expression.

Fig5, again additional contour label in a and b would be good. Please describe arrows in the caption.

[Figure]

Fig 7, it is hard to see the channel location here. Would elevation contours help here? Or something to clarify the location of the channel, which gets a bit lost especially in c,d. In fig 7c I am guessing that the red region is on the channel side and the blue region is the middle of the channel, but some visual clarification of this would be useful.

Given that channels appear to melt preferentially at the side, why do they not just keep getting wider? Is this balanced by lateral convergence of the ice flow? You mention lateral convergence in the context of data processing, but perhaps this should also be mentioned in your discussion of melting in sub-shelf channels. Could you view it as a competition between the ice dynamics trying to close these channels through lateral convergence and the ocean dynamics trying to open them through preferential melting at the side walls? Are there circumstances under which one would win over the other or are there feedbacks that prevent either from winning? Perhaps preferential side wall melting would steepen the side walls causing an increase in lateral convergence?

Did you think of using statistical techniques to investigate the relationship between LBMB and potentially relevant parameters such as spatial gradient of the ice shelf lower surface, absolute depth of the lower surface, distance from grounding line? This would shed some light on the relevance of currently used basal melt parameterisations (which are mostly linear functions of depth) in marine ice sheet modelling. Perhaps this would be a separate study, but really someone should be doing this urgently, and you have a good data set for it here! I feel sure that one could empirically justify a melt parameterisation as a function of both slope and depth more easily than just depth.

---

## Referee Comment (RC2) · G. Moholdt (Referee) · 3 Jun 2017

G. Moholdt (Referee)

geir.moholdt@npolar.no

Current mass losses from the Antarctic ice sheet are dominated by ice-shelf basal melting, but yet we know relatively little about the variability of basal melting and refreezing at the scale of individual ice shelves. This paper uses various high-resolution data sets from remote sensing to derive a detailed map of basal mass balance for the Roi Baudouin ice shelf in East Antarctica. The applied data sets and methods have several novel aspects, and the results are interesting in both a glaciological and oceanographic context.

The paper is well written and easy to follow. The methodology is well described, the

figures are clearly presented, and the discussion is straight to the point. I have a few general issues/questions and some smaller comments/edits as given below.

Lagrangian vs. Eulerian: I think the authors exaggerate about the "necessity of the Lagrangian approach" (P8, L15), at least if they mean it to be generally applicable. I agree that it is by far the best approach with the data sets they have at hand; i.e. two high-resolution DEMs that can probably be more accurately co-registered to each other (Lagrangian) than to an absolute reference system (Eulerian). But if consistent elevation and velocity data were available, there would be nothing in the way of getting reasonable Eulerian results. In fact, the authors fail to show that the Eulerian approach does not work in their case because they do not try to calculate Eulerian thickness changes. As long as the DEMs can be consistently georeferenced, it should not be much extra work to calculate and account for that to obtain real Eulerian BMB in Fig. 3. I do not doubt Lagrangian is better, but it would be nice to see it demonstrated as a comparison to the patterns in Fig. 1.

Ice shelf mass balance: The authors do not provide overall estimates of any mass balance components. That could be because the data sets do not cover the entire ice shelf or because they think that inherent biases are too large to do it confidently. However, I think that even some rough area-averaged estimates would be useful to include, or at the very least you should explain why this was not done and which challenges remain to be able to do it. Would flux gate methods be more reliable for that purpose? Potential biases between the 2013 and 2014 DEMs should be possible to correct quite well with CryoSat-2, whereas changes in firn air content and ocean properties are probably more difficult to assess.

Specific comments and questions in chronological order:

P1, L9: Is this range an estimate of actual min/max BMB or does it also contain impact from measurement noise? I guess that some erroneous values would be even larger.

P1, L11: Can be interpreted as if the radar profiling is an error source. Perhaps more

clear to say something like: "...although independent radar profiling show..."

P2, L1: has emerged

P2, L17: ice-sheet promontory

P2, L24: specify "several uncertain quantities" rather than just "large numbers"?

Fig. 1: I got a little bit confused about the letter labelling (a, b, c) and the panel labelling (a, b, etc.) in the later figures that they are connected with. I suggest to rather label the frames with the figure numbers they refer to (4, 5, 7) and then the three regions of interest with letters, like A, B, C.

P4, L1: Euelrian also requires two thickness fields in time to calculate dH/dt. In that sense, I do not see any difference with the Lagrangian approach. It is only the reference frame that is different (fixed or moving). See the general comment about this issue.

P4, L28: Any suitable reference to this technique?

P4, L24: Can the coverage of these three velocity datasets be indicated in Fig. 2c? That would be helpful for interpreting noise and smoothness in the divergence field in 2d.

P6, L13: Depoorter et al. (2013) is a composite grounding line from several other published ones. What is the real source in this case?

P6, L6: Do you use a steady-state firn air content or a time variable one? In any case, changes in firn air content (mainly due to accumulation anomalies) are a major uncertainty for the derived thickness changes because errors get incorrectly magnified by a factor 10 in the freeboard-to-thickness conversion. This should be mentioned.

P4, 13: This level of detail does not really fit here in a general description of the sections. I would rather describe the DEM differencing at the end of section 2.4 with a little bit more detail than here. The method and 5x5 km processing is not completely clear to me.

P6, L28: What about corrections for ocean tides and the inverse barometer effect?

P7, L11: Fig. 7 is a very nice illustration of this improvement, but is not shown until page 15. I think it should be moved forward here (Fig. 3?) since it helps to understand the purpose of the methodology.

P7, L27: Should be mentioned earlier together with the DEM methodology.

Fig. 3: I do not really see the relevance of this figure since we know that Eulerian elevation changes have a very variable pattern due to advocating topography. Assuming steady-state dH/dt is not a valid approach for determining spatial patterns of BMB, only area-averaged BMB. As mentioned earlier, I would rather like to see 2-3 panels with dH/dt from DEM differencing, maybe also u*div(H), and Eulerian BMB accounting for both of those contributions.

P8, L8-9: Label 2 and 3 are switched.

P9, L1: Why does it need to be "prescribed from external datasets"? Why not use your own DEMs?

P9, L3: The term steady-state is confusing here. I would rather highlight that it violates the ability to derive spatial patterns of basal melt.

P10, L2: Nice to be able to see this!

P10, L20: I would say opposite. Surface lowering is caused by negative LBMB.

P13, L34: I agree, it is better to use the same data directly than external sources.

P14, L5: This is a good novel approach. Well done.

P14, L27: I agree, but what about the channels? The effect of incomplete hydrostatic equilibrium across the channels is not discussed much (e.g. in relation to Fig. 4).

P14, L30: Is the gradient more important than the absolute thickness? The latter is not discussed, but is important for the pressure melting-point of the ice.

P15, L17: Reference Drews et al. 2017, Nature Comm.

P16, L14: Good point!

---

## Author Comment (AC1) · 11 Aug 2017

**Detecting high spatial variability of ice-shelf basal mass balance (Roi Baudouin ice shelf, Antarctica) S. Berger et al – TCD 2017 – doi:10.5194/tc-2017-41**

Here we respond point-by-point to the 2 reviews and the interactive comment. We found that the overall feedback is positive and constructive. We thank the referees David Shean and Geir Moholdt for their time and respond to their comments below. The main changes in the revised manuscript include:

- Reordering of the figures, addition of new Figures 1 and 9 and considerable modifications in new Figures 3 and 5.
- Inclusion of three supplementary Figures
- A more detailed accuracy assessment of the TanDEM-X DEMs
- The reorganisation of section 2 (data and methods) and 4 (Error sources). The former has been changed to better explain the calibration and matching of the different DEMs while the latter is reorganised to better emphasise our accuracy analysis of DEMs

In Eqs 1 and 2 we have replaced H with Hi to state more clearly that the thickness is computed and considered in ice-equivalent units. The main conclusions of the paper remain unaltered.

Our replies are printed in blue and numbered as G1.x and G2.x for general comments from reviewers 1 and 2, respectively. Specific comments are referred to with page and line numbers used by the reviewers.

Sophie Berger, Laboratoire de Glaciologie, Université Libre de Bruxelles, Belgium

**Response to Reviewer 1** (David Shean)**

**Summary**

This paper documents the high-resolution basal mass balance for the Roi Baudouin ice shelf for 2013-2014 using TanDEM-X DEMs. The authors provide a good description of the Lagrangian basal mass balance derivation (building on previous literature for analyses of other ice shelves), and compare DEM-derived basal mass balance with in situ pRES data and GPS-derived basal mass balance. In general, the methodology, results, and figures are well done. There are several issues that require further attention before publication. The authors claim to use a novel method to improve quality of velocity divergence products. The resulting products still contain fairly significant noise and the authors provide only one comparison with a coarse smoothing approach which generally looks similar. There are some significant discrepancies between firn air content from a dynamic firn densification model (RACMO-FDM) and the in situ radar data, with different assessments of significance. The authors offer a detailed analysis of an ice-shelf surface depression, which they interpret as a moving, unfrozen englacial lake 30 m below the surface. There is limited evidence to support this speculative interpretation. The LBMB derived from pRES/GPS measurements shows some significant differences when compared to the DEM derived LBMB. On the whole, this is a good paper that should be published after these and other issues are addressed.

We appreciate your detailed comments and have included many of them in the revised version. We first address the general comments before dealing with specific ones.

**General comments**

G1.1: The authors claim to use a novel method to improve quality of velocity divergence products. The resulting products still contain fairly significant noise and the authors provide only one comparison with a coarse smoothing approach which generally looks similar.

We agree that the initial version did not sufficiently discuss the differences between smoothing and total-variation regularization. Our initial motivation for using the latter was that it treats jump discontinuities more accurately than the corresponding smoothing approaches (Chartrand, 2011). This is important, because we suspect that ice-flow velocities change abruptly in ice-shelf channels that experience strong basal melting (Drews, 2015).

We now compare the regularized velocity divergence (with a fixed alpha given by the estimated variance in the velocity data), with divergences derived from velocity fields that have been smoothed to a varying degrees (average filter with kernel dimensions varying from 375x375 m to 1,875x1,875 m). The results are displayed in the new Fig 3. We find that the regularized derivative shows the strongest ice-flow divergence across an ice-shelf channel. Outside the ice-shelf channel, the regularized derivative is smoother than the derivative resulting from the 375x375 m smoothing, which is the only velocity field with a comparable divergence in the channel's surface depression. This shows that smoothing and regularization are not equivalent in detecting small-scale velocity anomalies around ice-shelf channels. However, for the Roi Baudouin Ice Shelf overall differences between regularizing and smoothing are relatively minor and the resulting LBMB estimates barely differ (because the correction for ice-flow divergence is small). This may not be the case for other ice shelves.

Figure 3 (new) has now been updated and we state this analysis now more clearly in the revised version.

G1.2 : There are some significant discrepancies between firn air content from a dynamic firn densification model (RACMO-FDM) and the in situ radar data, with different assessments of significance.

It is correct that RACMO-FDM does not correctly resolve the wind-albedo feedback close the grounding line. Consequently, the observed blue-ice area is not adequately represented in the modelled firn-air content. The impact of this misestimation on the derivation of the hydrostatic ice thickness has been discussed by Lenaerts et al. (2017). We reference this discussion in the revised version and mention the correspondingly increased uncertainty in the LBMB rates in those areas.

G1.3 : the authors offer a detailed analysis of an ice-shelf surface depression, which they interpret as a moving, unfrozen englacial lake 30 m below the surface. There is limited evidence to support this speculative interpretation.

Lenaerts et al (2017) witnessed subsurface meltwater features, such as the englacial lake shown below on the Roi Baudouin Ice Shelf. Using satellite data they identify 55 other and potentially refrozen englacial lakes on the ice shelf. The elliptical depression is one of them.

It is correct that we have no direct evidence for liquid water beneath the elliptical surface depression. It is now more clearly stated in the text, that if the elliptical depression once formed an englacial lake, it has probably refrozen by now and that the refrozen interface could block radar penetration.

This is now clarified in the text. We now only mention the englacial lake found farther upstream. The feature farther downstream is now referred to exclusively as an "elliptical surface depression" clearly marking that the origin of this feature is not fully known.

G1.4 : The LBMB derived from pRES/GPS measurements shows some significant differences when compared to the DEM derived LBMB.

We agree that the term near-perfect is too strong. However, given that estimated error of the LBMB is easily with a few meters per year we were surprised to see such little deviation on the eastern side

of the elliptical surface depression. The deviations on the western (i.e. left) side are certainly more significant and maybe linked to a more complex topography (finger-like features). We adapted the wording for comparing the pRES data with the LBMB accordingly.

**G1.5: abstract ending**

The abstract ends with a mention of challenges for full coupling between ice and ocean models, but this is never addressed in the paper.

Thanks for pointing this out. We find it not uncommon to end the abstract with a phrase stating the wider context, even though this context is not fully explored. To substantiate this point, we added a sentence in section 5 "Second, this sub-kilometre variability in ice-ocean processes poses challenges for coupling ice with ocean models, because highly resolved oceanic models have a typical resolution of 1-2 km (Dinniman et al, 2016), and community efforts such as the Marine Ice Sheet–Ocean Model Intercomparison Project prescribe horizontal gridding of 2 km (Asay-Davis et al, 2016). This is too coarse to fully capture the spatial variability that we observe here."

**G1.6 : structure of the method section**

Methods section needs some reorganization. Recommend separating DEM generation from hydrostatic ice thickness calculation. Agreed. We have now reorganized the old section "Hydrostatic ice thickness" (2.4-old) into two subsections "surface elevations" and "hydrostatic equilibrium" (not numbered). Moreover, we have added a new section "Lagrangian thickness change" (2.5-new). Here, we now explain the matching procedure in greater details.

**G1.7 : Analysis of TanDEM-X accuracy**

No meaningful analysis of TDM DEM accuracy is provided, only "estimated relative vertical accuracy better than 1 m (based on the standard deviation in overlapping areas)" – where are these areas? Are they all flat, or does this include higher surface slopes (like the surface depressions over channels discussed in the text)? Are you certain these areas are not evolving during the 2013-2014 period due to SMB or dynamics? A much more convincing approach would be to show a figure of DEM standard deviation over static surfaces (exposed bedrock, slow-moving ice with limited SMB).

Agreed. In the revised version we now address the DEM accuracy in the new subsection "Calibration and accuracy of TanDEM-X elevations" of section 4.1. To do so, we are using the following techniques:

1. Fig. S3 (Supplements) now shows the difference maps between overlapping individual frames from the same day, after calibration with a constant offset and Gaussian filtering but prior to the plane fitting. As detailed in the text, the offset accounts for unknown offsets from phase unwrapping, tidal uplift, inverse barometer effect, and for the spatially averaged depth penetration of the TanDEM-X signal. The applied offset correction is typically less than a few meters. Difference maps in new Fig. S3 show a discrepancy of 0.0+/-0.3 m with a slight spatial trend. We attribute these trends to residual phase ramps occurring during the bistatic processing of the TanDEM-X data. This bias typically ranges from -0.2 to 0.2 m except for the two northernmost difference fields which exhibit trends from -0.8 to 0.8 m. We correct these trends with plane fitting of 2014 to 2013 frames (Sect 2.4). In addition to the systematic bias, discrepancies with a magnitude of 0.5 m occur in steep areas (e.g ice-shelf channels or surface ridges).

2. New Fig. 9 compares the 2013 and 2014 TanDEM-X mosaics with two GNSS profiles.

• The mosaic from August and October 2013 is compared with a kinematic 20 x 25 km GNSS survey collected in December 2012. The investigated area includes strong topographic changes due to ice-shelf channels and a pinning point. The observed discrepancy is -0.44 +/- 1.05 m. Differences are largest inside ice-shelf channels because of dynamic topographic changes due to ice advection in the 8-10 month time interval between

GNSS surveying and DEM acquisition. Excluding these areas lowers the discrepancy to -0.37+/-0.29 m.

• The mosaic from June and July 2014 deviates with -0.04+/-0.65 m from a 100 km GNSS transect collected in December 2014. Similar to above, differences are largest in ice-shelf channels (i.e. the advection of an ice-shelf channel junction within the 6-7 month interval) and they lower to -0.07+/-0.55 m when ice-shelf channels are excluded from the comparison.

Apart from the channel, whose dynamic influence smear out other potential biases (changing SMB, variable penetration depth), we find the largest discrepancy close to the grounding line. There, the TanDEM-X elevations are overestimated by up to 2 m. This is discussed in reply G1.8.

The straightforward approach to evaluate our DEM at rock outcrops, or areas of zero SMB, is not possible because such areas do not exist in our area of interest. (Rock outcrops are extremely rare in all of the coastal Dronning Maud Land areas). Due to offset correction and plane fitting, we have to assume that the overall  $\partial$ Hi / $\partial$ t is zero.

All these points are now included in the revised manuscript in Section 4.1 and Fig S3 (supplementary) and Fig. 9

**G1.8 : Penetration of radar**

Are you convinced that you are not seeing penetration of the radar into snow and ice in the TandDEM-X DEMs? Other studies have shown this can be several meters, potentially up to 10 meters in cold, dry snow. This could impact the comparisons with GNSS surface elevations. Thanks for pointing this out. Humbert and Steinhage (2011) estimated that the TerraSAR-X signal penetrates in to the dry snow of the Fimbul Ice shelf by 8-10 m. In our case, the bulk part of such a signal penetration would be accounted for during the offset correction of the DEMs. However, a spatially or temporally varying signal penetration would result in a bias of our LBMB estimates.

In our area of interest, we would expect the largest spatial gradient in signal penetration in the North-South direction, where the depth of the firn column changes from about 15 m (firn-air content) near the ice-shelf edge, to 0 m in the blue ice belt close to the grounding line. As explained in the previous reply (G1.7), the TanDEM-X DEM in this area is systematically higher by about 2 m. This may be a relict of a variable signal penetration from the TanDEM-X satellites.

We discuss this point now more clearly (section 4.1), although we do not have the means to fully resolve it.

**G1.9 : combination of TanDEM-X frames**

No dates (month, day) are provided for the TDM DEMs, and we don't know over what period the "32 from 2013" and "11 from 2014" actually cover. Agreed. Figure S1 (supplementary) now shows the location of the different TanDEM-X frames and their acquisition date. In addition, section 2.4 has been rephrased to "The processing provides 43 single DEMs (32 from 2013 and 11 from 2014) gridded to 10 m. They cover a time span ranging from 21/06/2013 to 10/07/2014 (Fig. S1 – supplementary). The maximum time difference at overlapping areas between the 2013 and 2014 DEMs is 379 days."

Were mosaics generated for each year, centred on Jan 1 of the respective year? Did you

compute Lagrangian Dh/Dt for each pair of DEMs, or for the annual mosaics? The only DEMs that are mosaicked and treated as one piece are the different frames acquired on the same day and from the same satellite path. The Lagrangian DH/Dt was computed for each pair of overlapping DEMs, meaning that the time difference (Dt) varies from one pair to another. This was taken into account when calculating the corresponding melt rates in meters per year.

What about offsets between DEMs within a single year – if they are months apart, won't there be significant advection for areas flowing 300 m/yr? As we shift the 2013 DEMs forward, the Lagrangian framework reflects the 2014 geometry. And, as shown in Fig. S1 (supplementary), the 2014 DEMs span from 07/06/2014 to 10/07/2014, with a maximum time difference of 33 days between two overlapping DEMs of the same year. Even in areas flowing 300 m/a, this coincides with a shift of only 3 pixels (compared to a shift of 30 pixels after a year).

**G1.10: Gaussian filter**

The authors indicate that they smoothed their input 10-m posting TanDEM-X DEMs with a gaussian filter (no filter dimensions provided, but 7-sigma implies a large kernel). This inherently reduces the resolution of these DEMs, and will smooth edges of small-scale features like channels.

The Gaussian filter we chose has a standard deviation/coverage of 7 pixels (or 70 m) in either direction. This means that points lying within that distance are weighted with 0.68. At 14 pixels distance (or a radius of 140 m) the weight increases to 0.95.

The size of the Gaussian filter is chosen based on a comparison with a GNSS transect collected in 2012. We investigated standard deviations/coverages from 1-10 pixels and found that using 7 pixels minimises the standard deviation between GNSS and TanDEM-X surface elevation. As shown in Figure S2 (supplements) the applied smoothing does not affect the shape of the surface depressions linked to ice-shelf channels (with a typical width of 1-2 km).

**G1.11:** It is important to be clear that the DEMs are measuring surface elevation and the surface expression of features interpreted as basal channels. You are not directly measuring channel depth using surface elevation. Agreed. we have changed the text accordingly.

**G1.12:** Are there any airborne radar profiles (OIB, BAS, etc.) over this shelf? Seems like there must be something available. If so, you can use observed ice thickness and deviation from floatation thickness calculated from your surface elevations to estimate firn air content [Holland et al., 2011]. Our analysis around the elliptical surface depression includes a comparison between the hydrostatic ice thickness with ground-based radar. We infer a theoretical firn-air content (assuming hydrostatic equilibrium and absence of marine ice) which corresponds to the approach of Holland et al. (2011). There are no other airborne profiles in this area which could be used to do this analysis on a larger scale. (OIB has not surveyed this sector of Antarctica yet, and other airborne profiles have focused on gravimetry requiring a large flight height deteriorating the radar data).

**G1.13:** The ordering of the figures is a bit odd – the comparison showing divergence for regularization vs. smoothed velocities should be shown after describing the method, or early in the results. The LBMB figure (the main result) should come after the component figure. We rearranged the Figure order, also based on suggestions from Reviewer 2. Figure 1 is now a new figure that locates the on-site datasets.

**Specific comments (and additional general comments that came up for specific sections)**

Page 1:

Line 12: Is the lake 30 m or the surface depression is 30 m below surrounding surfaces? We deduced from the radar profile that the upper surface of the characteristic radar reflector lies approximately 30 m below the surface. However, the depression is 10 m lower that the surrounding., and this is the value that we now mention in the abstract.

Lines 16-17: see reply G1.5

Line 21: Not sure "Marine Ice Sheet Instability" should be capitalized, changed to 'marine ice sheet instability'

Page 2:

Line 10: delete "the" before "the BMB" We are not sure about the correct usage here and will keep

**an eye on it during typesetting.**

Line 11: "which form" done

Line 11-12: I don't understand this 50% claim, also, the sentence ends abruptly – missing "surrounding ice"? We have changed this sentence to make it more understandable: "Ice-shelf channels are one expression of localised basal melting (Stanton et al, 2013,Marsh et al 2016) which, due to hydrostatic adjustment, form curvilinear depressions visible at the ice-shelf surface (Fig. 1). These surface depressions virtually always reflect basal incisions resulting in curvilinear tracts of thin ice. In some areas, ice-shelf channels are twice as thin as their surroundings (Drews, 2015)". Note that an 's' was missing in surroundings.

Line 10: What about GPS receivers [Jenkins et al., 2006; Shean et al., 2017]? Reference to Shean et al (2017) has been added.

Line 14: Channel carving does not cause increased crevassing – these are separate processes. The link between ice-shelf channel formation and basal crevassing has been proposed by Vaughan et al (2012) (see figure below).

c. Zones of possible failure

Figure 6. Drawing of the formation of surface and basal crevasses based on thin-beam theory.

Line 16: delete "at 10 m gridding" – not relevant for intro Thank you for your suggestion but we would like to keep this part, as the spatial resolution is important for the paper.

Line 17 and later: I believe TC should be "Figure" not "Fig." Journal guidelines require using "Fig." unless at the beginning of a sentence where "Figure" should be used. We have adapted our notation accordingly.

Line 22: delete "all the way" done

Line 23: delete "In the following," done

Line 24: delete "results", also suggest "with focus on..." rather than "special focus" done

Line 26-28: "accounts for" is a bit awkward – you are mitigating the noise, not accounting for it agreed. Changed to "mitigates the noise"

Line 29: change "observational evidence" to "observations" done

Line 29: suggest simplifying to "...validated with phase-sensitive radar, ground-penetrating radar, and GNSS observations." done, we have rephrased your suggestion "validated with phasesensitive radar, GNSS observations and ground-penetrating radar."

Page 3:

Equation 1: This is 3 equations! Should probably split. You might start with the standard mass conservation rather than rearranging at the start. Thanks for the suggestion. The three lines are now referred to as sub-equations (1a), (1b) and (1c).

Line 6: "vertically integrated" (no hyphen needed) – perhaps "column-average" is a better descriptor here. You are assuming surface velocity is equal to column-average velocity, this should be stated somewhere. Agreed. Changed to "the column-average horizontal velocity of the ice". We also added this at the beginning of section 2.2 (surface velocities from satellite radar remote sensing) : "Assuming that that velocities do not vary with depth....." Line 6: "In principle, " done

**Page 4**

Line 1: Not sure what you mean here - Eulerian dH/dt involves two thickness measurements. What we mean is that usually studies using an Eulerian framework only rely on 1 thickness field and an external dataset for the thickness change, instead of computing it from two different thickness fields. We have now rephrased that part: "Eulerian studies are often based one thickness field and either assume steady-state (Rignot and Steffen, 2008; Neckel et al 2012) or rely on an external dataset (Depoorter et al, 2013, Rignot et al, 2013) to account for the thickness changes  $\partial$ Hi / $\partial$ t (e.g. Pritchard et al, 2012; Paolo et al 2015) ".

Line 6: "can only be adequately done" is subjective. Agreed. Changed to "The Lagrangian approach is best-suited in areas where advection is significant (e.g. near ice-shelf channels)."

Line 7: Add [Shean, 2016; Shean et al., 2017] for additional derivation and discussion of Eul/Lag elevation change. Shean et al, 2017 added. Unfortunately we do not have access to the PhD thesis Shean (2016).

Line 10: "from 2013 and 2014" done

Line 10: Suggest deleting ", clearly resolving ice-shelf channels" – you are claiming that the 10-m resolution is novel (a claim some might reject). This resolution can resolve many small-scale features on ice shelves, not just channels. OK, removed.

Why so many details in paragraph 2? Details like "we calculate Lag thickness change by crosscorrelating the TanDEM-X DEMs (using 5x5 km2 patches...) should be in Section 2.5. Agreed. We have added a new subsection "Lagrangian thickness change" (2.5-new) (see response G1.6)

5x5 km2 – I think you mean 5 km x 5 km. yes changed.

OK, so you have 10-m DEMs, and you are cross-correlating using a 500x500 pixel kernel. Using a kernel this large inherently reduces the resolution of the output velocity products. Did you do this for every pixel, or for some sparse interval? We think we did not explain this part very well: the velocity field is not a product from the DEM matching. Instead we matched the DEMs and then used an external velocity field of the area (from Berger et al 2016). We matched DEM patches of 5x5 km sampled every km (to make sure to the shifting covers every area). This is now better explained in the new section "Lagrangian thickness change" (2.5-new).

Line 17: "freely floating" Sorry we don't understand that comment.

"viscous inflow in ice-shelf channels" is a process, not a "small-scale feature" Ok, we rephrased the sentence "...but also other small-scale features such as ice-shelf channels where viscous inflow can occur ..."

Line 24: "of 1996" and "of 2010" should be "from 1996" done

Line 24-25: So, you compared the 1996, 2010 and 2014 velocities and found no evidence for temporal variations? What are dates of the input InSAR? What was time period of GNSS? Surely

**the velocities aren't identical. The remote sensing flow field is a mix of InSAR velocities from 1996 and speckle tracking from 2010 (see table below from Berger et al (2016), for more details)**

**Table 1.** Characteristics of the satellite data;  $\Delta T$ ,  $\lambda$  and  $B_{\perp}$  are the temporal baseline, the wavelength of the sensor and the perpendicular spatial baseline between the master and slave images, respectively. The satellite frames are shown in Figure 1

| Processing       | Sensor      | $\Delta T$ | λ   | Track | Date (master)     | $B_{\perp}$ | Orbit      |
|------------------|-------------|------------|-----|-------|-------------------|-------------|------------|
|                  |             | d          | cm  |       |                   | m           |            |
| InSAR            | ERS 1/2     | 1          | 5.6 | 320   | 21 May 1996       | 37          | Descending |
|                  |             |            |     | 430   | 28 May 1996       | 62          | Ascending  |
| Speckle tracking | ALOS-PALSAR | 46         | 23  | 661   | 1 August 2010     | 520         | Ascending  |
|                  |             |            |     | 661   | 16 September 2010 | 437         | Ascending  |
|                  |             |            |     | 661   | 1 November 2010   | 453         | Ascending  |
|                  |             |            |     | 665   | 8 October 2010    | 588         | Ascending  |

The yearly GNSS-velocities were acquired over several consecutive field campaigns, either December 2012-December 2013 or December 2013-December 2014, with a time span ranging between 362 and 368 days. Some of the GNSS velocity points served to calibrate the satellite-based flow fields and the others were used as control points.

Berger et al (2016) also compare the satellite-based flow field with 74 completely independent ground-truth measurements collected in 1965–67 (Derwael, 2014) and infer that "*The deviations are not larger than the uncertainty of our high-resolution flow field and we conclude that the RBIS has not undergone prominent changes in average ice flow over the last five decades.*"

To clarify, we have rephrased the sentence as "As shown in Berger et al (2016), comparison with on-site measurements collected in 1965-1967 and 2012-2014 yields no evidence of prominent changes in the ice velocities over the last decades, which supports the combination of data from different dates."

Line 25: "This dataset" – ambiguous agreed. We changed to "this velocity mosaic" (specifying before that the velocities have been mosaicked)

Line 27: Replace "cutting edges" with "seams" done

"Offsets between the two datasets are over 60 ma-1 in places." You are blending seams with feathering, but you are not actually correcting the offsets in velocity magnitude. The velocity mismatch at the seams varies from one place to another and can reach up to 60 m/a. As a result, it is impossible to remove the seams with a constant offset.

This will lead to smooth gradients for the divergence, but will likely lead to incorrect horizontal path determination for your Dh/Dt. That 60 ma-1 is a significant portion of the ~50-300 ma-1 velocities over the ice shelf. Your Lag Dh/Dt obs could be "off" by 6 pixels at your DEM resolution. The velocities are exclusively used to compute the divergence term. Also note, that the 60 m/a is a maximum deviation, the mean and standard deviation are much smaller . The DEMs are shifted with a normalised 2D cross-correlation. Shifting the DEMs with the velocities was less reliable because, as we explain in section 4.1, the velocity direction is not sufficiently constrained.

Line 30: "The SMB is based on…" is awkward start. You are using RACMO, not basing your SMB on RACMO. Good point, we changed to "We use the surface mass balance from a high-resolution (5.5 km posting) simulation of the Regional Atmospheric Climate MOdel (RACMO)…" Line 32: Need a reference for these processes that RACMO is reproducing. We now refer to Lenaerts et al (2014)

Also, RACMO doesn't predict anything. 'predict' has been changed to "simulates" Add reference to Figure 2b when discussing SMB spatial distribution. done

Section 2.4: This section should be split into 1) DEM processing and correction and 2) Hydrostatic thickness calculation. We have divided this section in 2 subsections : surface elevations and

hydrostatic equilibrium.

Switching between "were" and "are" – should all be past tense To be consistent, we put everything in present tense

Add reference to TanDEM-X mission. Krieger et al (2007) added

Add reference for SARscape software? There is no peer-reviewed reference for the software that we are aware of but we have added a ® to indicate that this is a commercial software.

How did you co-register the DEMs? How do we know there aren't horizontal and vertical

offsets between your input DEM data that will lead to artificial elevation change (and LBMB)

**signals?** As explained in subSection "Hydrostatic thickness", we use a 2D cross-correlation to corregister DEM. The 2013 DEMs are tied with a simple offset to a CryoSat-2 DEM (Helm et al, 2014) and the 2014 DEMs are adjusted to the 2013 DEMs by fitting a plane, to correct for linear trends visible in the 2013-2014 difference fields. To do so, we have to assume that the absolute change in surface elevation between the two years is small.

Looks like a seam artifact is present in Figure 1 between labels 3 and B. LBMB values are positive (~3-5 m/yr) and then immediately adjacent, close to 0. Yes indeed, the LBMB data change rapidly over short distances because the value from 2 different frames (processed independently) are overlain. Frames boundaries are now visible in Fig S1 (Supplementary).

Line 12: Don't start sentence with acronym agreed. Changed to "Digital elevation models" where it first appears then to "elevations"

Line 11: Based on my experience with Greenland data, I am skeptical of this 1 m vertical accuracy for TDM. This needs stronger justification. See reply G1.7

Line 18: What are dimensions of your Gaussian filter – 7 sigma is large, implying a large kernel,

which will significantly reduce the resolving power of the output DEM (definitely not 10-m) See reply G1.10.

Weren't the DEMs and GNSS data collected during different time periods? Are you assuming that no change occurred between the two collections? Yes we have to assume little change in the 6-10 -month period.

What is the "mean and standard deviation" of the differences? We subtracted the TanDEM-X elevations from the GNSS values and calculated the mean and standard deviation of subtraction. The GNSS data are isolated to a small area, how do we know that this is representative of the larger ice shelf? The GNSS data used to determine the size of the filter are acquired over a square of ~20x25 km with abrupt topographic changes (Fig 1 – new). We use it to ensure that the filter does not smear out small scale details. Moreover we now also include a 100-km long, north-south oriented transect that covers 75 % of the along-flow extent of Roi Baudouin Ice Shelf to discuss the accuracy of the DEMs (reply G1.7)

Line 22: Wording is awkward, suggest something like "We calculate freeboard ice thickness assuming hydrostatic equilibrium..." Changed to "We invert the hydrostatic thickness from freeboard heights"

Line 25: Firn air content accounts for total air content of the firn, not variable firn density. To be clearer, we have rephrased the text : "The firn-air content Ha accounts for the lower firn and snow densities by subdividing the ice column...."

What are typical mean dynamic topography offsets for this location? In our area of interest, the mean dynamic topography ranges from -0.9 to 0.6 m (with an average of -0.1 m). This is now specified in Table 1

Are you accounting for density errors in your uncertainty estimates? I've found that including a

+/- 5 kg/m3 uncertainty in density can dominate the freeboard thickness error, much more than a few meters of firn air content error. This comment is a little unclear. Uncertainties in firn-air content do implicitly mean errors in the (depth-averaged) density. Do you refer to the assumed ice density (e.g. 900 vs. 910 vs. 917) ? For a thickness of about 300 m, changing the assumed density of ice by 5 kg/m3 has the same effect as changing the firn-air content by about 1 m (both cause 6-7 meters thickness change in the hydrostatic ice thickness). The uncertainties due to density are discussed in section 4.

I've found that IMAU-FDM estimates over ice shelves in West Antarctica are biased high, in some places 5-10 m too high, compared to radar-derived firn air content using techniques from In our case, the firn-air content from IMAU-FDM deviates with 0.7+/-2.0 m (with a maximum deviation 3m) of from radar-derived firn air content in Drews et al (2016), who analysed 5 wide-angle surveys, including 1 data point in an ice-shelf channel where the results were verified with ice-core data.

Review was interrupted for several weeks at this point. Picking up again. I apologize for discontinuity in comments.

Line 28: What is approximate mean dynamic topography correction for this location? See reply p6, L25

**Page 7**

Line 5: I think you mean "these approaches are not well-suited..." True but the whole sentence has now been rephrased to be more specific. "However, smoothing prior to taking the derivative can lead to smearing out of the derivative in areas where the derived quantity changes abruptly (or discontinuously)."

Line 9: What is meant by "wiggliness" of the derivative? Need a little more explanation about what you are solving for and how the process works. This is emphasized as a novel method, so it should be documented clearly. Agreed. The wiggliness refers to the second derivative which becomes very large if the data are noisy (this because the finite difference schemes pick up the local slope of the noise as opposed to the larger scale signal). We have rephrased section 2.5 (including a comparison with smoothing the data to different degrees). See also reply G1.1.

So, you are computing horizontal and vertical gradients separately, then combining in a large inversion? We use the regularized derivative separately for x and y and then combined them

How does the velocity map resolution impact alpha? The map resolution indirectly influences alpha in the sense that lower resolved velocities (i.e. spatially averaged and less noisy) require less regularization when taking the derivative and vice versa.

When I look at Figure 2D, I still see plenty of noise in the velocity divergence map. We have justified our choice of alpha (i.e. the regularization). As prescribed by (Chatrand, 2011) we use the discrepancy principle to choose our alpha, i.e. we rely on the estimated noise in the velocity field. Line 18: replace "such as" with "including" done

OK, so you de-tided the GPS surface elevation data. Yes. Did you also detide the TDX DEMs? Yes, indirectly by calibrating the TanDEM-X DEM (with the grounded parts masked out) to a de-tided DEM

What depths were the reflectors used to determine strain thinning? Did you have to account for firn compaction? The strain thinning is based the bottom of the ice shelf and upper reflectors located 60 to 90 meters below the surface. Therefore, we don't need to account for firn compaction.

Page 8:

Lines 1-2. I don't understand this. You are saying the strain correction is small compared to the basal melt rate, with both provided in units of m/yr. This makes sense. The 10-day interval should be irrelevant here – why is it mentioned? Thanks for pointing this out, the strain rate should have units per year (not meters per year) and we have removed the part about the 10-day interval. Also correcting for strain thinning is standard in pRES processing (also when it is small).

This result about strain correction suggests that the velocity divergence term (and the regularization) is not necessarily important for the larger shelf LBMB calculation. The Correct. We

have mentioned that in the old version on p13 L30-31 (old ) "Fortunately, for the Roi Baudouin Ice Shelf this effect is mitigated by the low ice-flow divergence,[...] This may be different for other ice shelves." The pRES data are in an area with low divergence.

Line 6: "seaward" done

In Figure 1, what is the band of large positive (blue) values along the grounding line to the right of Label "1"? Artifacts or real refreezing signal? My guess is that the Depoorter grounding line is in the wrong place, the ice at this location is grounded, and this area should be masked.

We don't know what is causing this band. A mislocation of the grounding line seems unlikely, as we have checked numerous grounding lines in this area, many of which coincide with ground-based measurements and show little temporal variation (Drews et al., 2017). Potentially the band is linked to unaccounted surface processes such as melt water formation which can be abundant in this area (Lenaerts et al., 2017).

Line 8: "stoss" is a relative term – could be leeward for ocean circulation, different direction for wind, different direction for ice flow – suggest changing to absolute direction ("south") agreed. Changed to "southern"

Line 11: Where are these overlapping areas? Are you sure that the LBMB is not changing over the period for which you are performing the analysis (could some of your observed std be due to real changes in melt rates?) I think you are saying that formal error estimates are larger than the magnitude of the measured signal.

The overlapping areas of the LBMB coincide with the areas where 2013 and 2014 TanDEM-X frames overlap with other frames of the same year but different dates. This is now shown in Fig S1 (supplementary). We have no handle on the temporal evolution of the LBMB, but all the TanDEm-X DEMs have been acquired during Austral winter. It is correct that our error estimates are larger than the signal. Therefore, we use these difference fields in LBMB as a lower boundary for our error estimate.

The last sentence on Page 8 and first sentence on Page 9 have no real context. I think you are making an argument that dh/dt from sparse or low-res measurements is problematic. But you have high-res DEMs, so you don't need external datasets for high-res Eulerian elevation change. Agreed, this section has now been rephrased and the part about "the need for external datasets" removed.

**Page 9**

Line 7: Is this order of magnitude difference present everywhere? Seems like DH/Dt values are close to 0 in the middle of the shelf, so the vdiv and smb terms become much more important. Good point. Overall the DH/DT term is the most important one. In some areas (where this term is close to zero) other terms are equally important but in those areas the LBMB signal is typically very small and close to the detection limit.

Line 11: Is this convergence within channels present across the full channel width, or just on the sides, or do you lack the resolution to determine this? This convergence pattern appears clearly in modelling studies (most prominently in the flanks), but is hard to pick up in observed surface velocity fields.

I don't see negative Dh/Dt across all channels in Figure 2f. In fact, I see positive Dh/Dt in several places (e.g., just northwest of the right-most arrow in figure 2d)." So, the convergence is causing thickening of the ice shelf at this location? More likely is that you are picking up surface elevation change due to snow redistribution.

It is correct that not all ice-shelf channels show the same pattern. However, some do and we link those to the modelling results of Drews et al., (2015). It is entirely possible that we also pick up some local changes of the SMB (as stated in Section 4.3 - old). We included that point in the revised manuscript.

Line 3: At the channel center, LBMB values are positive, potentially even +2-3 m/yr. This is not close to zero. What is going on here? Is that refreezing (unlikely) or is this an artifact? Need to address this if you are going to interpret the signals on the sides with confidence.

The absolute LBMB values are inflicted with an error that is larger than the signal. The key observations here is therefore that the absolute magnitude of the LBMB is 3 times higher in the flanks compared to the channel's centre (-5 m/a vs 1.5 m). Our emphasise is much more on the spatial variability than on the absolute values. Because enhanced melting at the channel's flanks has been suggested for wider ice-shelf channels (Millgate et al, 2013), we find this observation meaningful even if uncertainties remain.

Line 4: not km2 here, just km done

Line 4: elliptical, not ellipsoidal replaced there and elsewhere in the text

Line 5: You don't necessarily know that the lake is connected. It's location is adjacent to the channel, but careful about wording that could be misinterpreted here. Changed to "elliptical surface depression [...] located on the upstream end of an ice-shelf channel"

Line 6-7: This interpretation about tributaries needs more support if it is going to be included. We have now replace the term tributaries with "fingers", which is used to describe the location and shape of the features feeding into the elliptical surface depression. We don't do any interpretation beyond that.

I haven't seen the Lenaerts et al (2017) paper, but I'm puzzled by this interpretation. Are you suggesting that the lake is liquid water, 30 m below the surface of the ice shelf? Why have these not refrozen? As explained in reply G1.3, we have made it much clearer here that there are multiple options that could explain this feature but that if it was a lake it has probably refrozen.

Line 12: "blocks the penetration" suggest "attenuates" We mean blocking in the sense of total reflection and negligible transmission.

This interpretation is inconsistent with the radar data shown in Figure 6. There are many reflections beneath the "lake" feature. This to me suggests there is no way that this is liquid water. There may be an interface that is attenuating the radar signal, but definitely not salty water, which seems like the only way to prevent refreezing. As stated above (p10 L6-7), and in the initial manuscript, we also consider that the interface is a refrozen (formerly liquid) lake. The revised version is correspondingly adapted to make this more clear. There are no radar signals originating from beneath the prominent interface (we have checked this with phase sensitive radar). The signals that occur at larger depths in Fig 8a(new) originate from off angle reflections from the lateral walls.

Line 15: I disagree – this interpretation is important. I would not call it an englacial lake if you have no direct evidence for this interpretation. Stick with "elliptical surface depression" and be consistent throughout. Agreed. See reply G1.3

I am not entirely convinced that the apparent LBMB over this feature is not due to variable snow accumulation and redistribution over the periods when you have elevation measurements. True, changes in SMB could, in theory, impact our LBMB. However, previous studies suggest higher accumulation in surface depressions (Langley et al, 2014) and at the bottom of slope (Frezzotti et al, 2007). Increased SMB at this location cannot explain surface lowering as it would decrease the inferred melt rates. As a result, unaccounted SMB variability would make this signal even more prominent. We don't see a mechanism why snow would be redistributed (by wind) out of the surface depression.

I disagree with the interpretation that the pRES and DEM-derived LBMB values agree "well" or show a "near-perfect" fit. Figure 6b shows major disagreement (+/-2-6 m/yr) between the two in places. These offsets are large and significant compared to the magnitude of the LBMB signal. See reply G1.4

This paragraph is very long, and should be broken up Done

**Line 19: I don't think "low" is the right term here, try "large negative" done**

In Fig 6c, I don't understand why the surface is getting lower over the depression, but getting higher between 0.5-2.0 km. Actually, the surface is also getting lower between 0.5-2 km (the black line is the oldest profile). So, if I understand correctly, the P-P' profile was extracted in a fixed Eulerian 2016 location? So some/all of the observed elevation change for this fixed profile could be due to advection? Why was not extract the profile in a Lagrangian sense, moving with the feature? The profile shown on this figure is already in a Lagrangian framework, i.e. the TanDEM-X DEMs have been moved forward to match the acquisition geometry of the 2016 GNSS profile. As a result, we don't think that the observed elevation changes are due to advection.

Are you convinced that you are not seeing penetration of the radar into snow and ice in the TandDEM-X DEMs? Other studies have shown this can be several meters, potentially up to 10 meters in cold, dry snow. This could impact the comparisons with GNSS surface elevations. Yes , see reply G1.8

It looks like the large negative values in LBMB along the channels is mostly coming from the velocity divergence term. Are you confident that these negative values on channel sides are not artifacts velocity resolution and regularization approach are a [word missing, we do not understand the end of the sentence] The three figures below present the different term influencing the LBMB in Eq (1) but unlike Fig. 2, in the paper, they are presented with the same colorscale. Looking at these maps, gives us confidence that the negative LBMB values on channel's sides are driven by the DH/Dt more than by the regularization.